



# Understanding the vertical temperature structure of recent record-shattering heatwaves

Belinda Hotz[1], Lukas Papritz[1], and Matthias Röthlisberger[1]

[1]Institute for Atmospheric and Climate Science, ETH Zürich, Zürich, Switzerland

**Correspondence:** Matthias Röthlisberger (matthias.roethlisberger@env.ethz.ch)

**Abstract.**

Extreme heatwaves are one of the most impactful natural hazards, posing risks to human health, infrastructure and ecosystems. Recent theoretical as well as observational studies suggested that the vertical temperature structure during heatwaves limits the magnitude of near-surface heat through convective instability. In this study, we thus examine in detail the vertical
temperature structure during three recent record-shattering heatwaves, the Pacific Northwest (PNW) heatwave in 2021, the Western Russia (RU) heatwave in 2010, and the West European and UK (UK) heatwave in 2022 by decomposing temperature anomalies ($T'$) in the entire tropospheric column above the surface into contributions from advection, adiabatic warming and cooling, and diabatic processes.

All three heatwaves exhibited bottom-heavy yet vertically deep positive $T'$ extending throughout the troposphere. Impor-
tantly, though, the $T'$ magnitude and the underlying physical processes vary greatly in the vertical within each heatwave, as well as across distinct heatwaves, reflecting the diverse synoptic storylines of these events. The PNW heatwave was strongly influenced by an upstream cyclone and an associated warm conveyor belt, which amplified an extreme quasi-stationary ridge and generated substantial mid- to upper-tropospheric positive $T'$ through advection and diabatic heating. In some contrast, positive upper-tropospheric $T'$ during the RU heatwave was caused by advection, while during the UK heatwave, it exhibited mod-
est positive diabatic contributions from upstream latent heating only during the early phase of the respective ridge. Adiabatic warming notably contributed positively to lower tropospheric $T'$ in all three heatwaves, but only in the lowermost 200–300 hPa. Near the surface, all three processes contributed positively to $T'$ in the PNW and RU heatwave, while near-surface diabatic $T'$ was negligible during the UK heatwave. Moreover, there is clear evidence for an amplification and downward propagation of adiabatic $T'$ during the PNW and UK heatwaves, whereby the maximum near-surface $T'$ coincided with the arrival of max-
imum adiabatic $T'$ in the boundary layer. Additionally, the widespread "ageing" of near-surface $T'$ over the course of these events now quantitatively supports the notion of heat domes, within which air recirculates and accumulates heat.

Our results for the first time document the four-dimensional functioning of anticyclone–heatwave couplets in terms of advection, adiabatic and diabatic cooling or warming, and shed light on the complex interplay between large-scale dynamics, moist convection and boundary layer processes that ultimately determines near-surface temperatures during heatwaves.



## 1 Introduction

Exceptionally intense heatwaves such as the infamous western Russian heatwave in 2010 (Barriopedro et al., 2011, hereafter "RU heatwave") or the recent heatwaves in June 2021 in the Pacific Northwest (Philip et al., 2022; Neal et al., 2022; Schumacher et al., 2022; White et al., 2023, hereafter "PNW heatwave"), and in July 2022 in western Europe (hereafter "UK heatwave") shattered local temperature records (Fig. 1), caused hundreds to thousands of fatalities, and significantly impaired a wide range

of ecosystems (White et al., 2023; Hermann et al., 2023; Vicedo-Cabrera et al., 2023). In light of these impacts, there are large societal and political demands for accurate projections of heatwave characteristics, particularly concerning such exceptionally intense events (Barriopedro et al., 2023). However, the reliability of heatwave projections ultimately hinges upon a physically accurate representation of these events in models, and assessing modelling capabilities in these regards in turn requires a detailed understanding of the underlying physical processes (Vautard et al., 2013). Moreover, such process understanding is

fundamental for constructing plausible storylines of future exceptionally intense heatwaves (Shepherd et al., 2018; Wehrli et al., 2019).

The proximal causes of "benign" heatwaves over mid-latitude land regions are clear and have been elucidated by a plethora of studies: Such events typically occur within anticyclonic flows embedded in a slow-moving larger-scale Rossby wave structure, e.g., an atmospheric block (Xoplaki et al., 2003; Meehl and Tebaldi, 2004; Stefanon et al., 2012; Pfahl and Wernli, 2012)

or a stationary subtropical ridge (Sousa et al., 2018). The formation of the unusually large positive near-surface temperature anomalies that constitute a heatwave occurs due to three processes: (a) In the upstream part of the anticyclone, air is advected across climatological temperature gradients from climatologically warmer to colder regions. (b) In the central part of the anticyclone, air typically experiences large-scale subsidence, which yields adiabatic warming and leads to clear skies, which (c) enhances the diabatic heating of near-surface air due to strong insolation and resulting sensible heat fluxes (e.g., Fischer et al.,

2007; Miralles et al., 2014; Bieli et al., 2015; Zschenderlein et al., 2019; Barriopedro et al., 2023). Moreover, the persistence of the large-scale anticyclonic flow, e.g., due to an upper-level blocking flow or recurrent Rossby wave pattern (Röthlisberger et al., 2019), also increases the persistence of the warm and dry surface conditions (Röthlisberger and Martius, 2019), leaving enough time for soils to dry out and for further amplifying the near-surface heat through land–atmosphere interactions (Fischer et al., 2007; Miralles et al., 2014, 2019).

However, it is far less clear what factors discriminate heatwaves with exceptional magnitudes from less intense events. Case studies focusing on the events mentioned above have confirmed the importance of anticyclonic flow patterns, air mass advection, adiabatic warming, clear skies and dry soils for the existence of these heatwaves (e.g., Dole et al., 2011; Schneidereit et al., 2012; Philip et al., 2022; Faranda et al., 2023), but the causes of their exceptional magnitude remain an area of active research. For the RU and PNW heatwaves, several studies provided evidence that the atmospheric vertical structure (in partic-

ular the vertical temperature structure) during these events was pivotal for determining the magnitude of the near-surface heat. Neal et al. (2022), Schumacher et al. (2022) and Oertel et al. (2023) documented the exceptional mid- to upper-tropospheric warmth during the PNW heatwave, and in particular Neal et al. (2022) and Schumacher et al. (2022) argued that these positive temperature anomalies aloft suppressed convective damping of near-surface temperature anomalies. Along a similar line of





reasoning, Zhang and Boos (2023) recently developed a theory for an upper-bound of near-surface temperatures during heat-
waves over extratropical land, which is based on the assumption that near-surface temperatures are limited by the stability of
the atmospheric profile to moist convection. The authors were able to demonstrate that their theory holds remarkably well for
near-surface temperatures and atmospheric profiles from reanalysis data. For the RU heatwave, Miralles et al. (2014) empha-
sised the importance of the atmospheric vertical structure by revealing that air, diabatically heated during the day and residing
in a nocturnal residual layer (far removed from the surface), re-entered the boundary layer on the following day. This diurnal
and vertically organised heat accumulation was found to be a pivotal factor in reaching exceptional near-surface temperatures
during this event.

In summary, there is accumulating case study evidence (Miralles et al., 2014; Neal et al., 2022; Schumacher et al., 2022) and
a theoretical underpinning (Zhang and Boos, 2023) suggesting that the atmospheric vertical structure during heatwaves, in par-
ticular the vertical temperature profile, is key for determining the magnitude of exceptionally intense heatwaves. This study now
specifically investigates how the vertical temperature structure during the recent record-breaking PNW, RU, and UK heatwaves
formed. That is, we quantitatively examine how the interplay between air mass advection across climatological temperature
gradients, adiabatic warming/cooling, and diabatic processes shaped the vertical temperature anomaly ($T'$) profile during these
events. As a main analysis tool, we use the Lagrangian $T'$ decomposition of Röthlisberger and Papritz (2023a), which is based
on kinematic backward trajectories and allows decomposing any $T'$ of interest into contributions from horizontal air mass
advection across climatological $T$ gradients, adiabatic warming/cooling, diabatic processes and a usually small residual. The
Lagrangian $T'$ decomposition has so far only been applied to near-surface $T'$ (Röthlisberger and Papritz, 2023a, b). Here, we
extend these studies by applying it to the entire tropospheric column above the PNW, RU, and UK heatwaves.

Hereafter, we introduce the data used in this study and then provide a brief introduction to the Lagrangian $T'$ decomposition
(Section 2). In Section 3, we discuss in detail the characteristics and causes of the atmospheric vertical structure during the
PNW heatwave and then contrast it with the atmospheric vertical structure during the Russian and European heatwaves. Finally,
we present our conclusions in Section 4.

## 2 Data and methods

### 2.1 Data

This study uses the European Centre for Medium-Range Weather Forecasts (ECMWF) ERA5 reanalysis (Hersbach et al., 2020)
at 3-hourly temporal resolution. Spatially, the data have been interpolated to a resolution of $0.5°$ latitude by $0.5°$ longitude and
vertically to a stack of pressure levels (from $1000\,\text{hPa}$ to $140\,\text{hPa}$ in intervals of $20\,\text{hPa}$). Note, however, that for Lagrangian
analyses (see below), we use ERA5 data on the original model levels. The regions of the three heatwaves for which data have
been analysed are listed in Table 1. The following ERA5 variables are used: temperature $T$, potential temperature $\theta$, wind
$\boldsymbol{u} = (u, v, \omega)$, specific humidity $q$, geopotential height $Z$, pressure $p$, mean sea-level pressure (SLP), potential vorticity (PV),
and the height of the planetary boundary layer (PBL). To define temperature anomalies $T'$, we use exactly the same temperature
climatology $\overline{T}$ (computed on model levels) as Röthlisberger and Papritz (2023a), which is transient and takes into account both





the diurnal and seasonal cycles. Specifically, $\overline{T}$ is computed for a given time step by averaging model level $T$ across all time steps with the same time of the day within 21-day and 9-year windows centred on the time step of interest. That is, each $\overline{T}$ value is the average of $21 \times 9 = 189$ instantaneous $T$ values at each model level. Wherever daily averages are presented, they

refer to averages over UTC (and not local) days, i.e., comprising 8 time steps from 00 to 21 UTC.

## 2.2   Stability to moist convection

Zhang and Boos (2023) convincingly argued that the largest possible magnitude of a heatwave is limited by the stability of the atmospheric vertical structure to moist convection. That is, near-surface temperatures can only rise until the atmospheric profile becomes unstable to moist convection, which would cool the surface through convective overturning and precipitation.

This argument provides a key motivation for studying the atmospheric vertical structure during heatwaves, in particular in cases where indeed the atmospheric vertical structure is near neutral to moist convection (i.e., near moist adiabatic). Here, we quantify the stability of the atmospheric vertical structure to moist convection during our three events of interest by following exactly the reasoning put forward in Zhang and Boos (2023). Specifically, we compute and compare the moist static energy at the surface ($\mathrm{MSE}_s$) as well as the saturation moist static energy at 500 hPa ($\mathrm{MSE}_{500}^*$) as

$$\mathrm{MSE}_s = c_p T_s + L_v q_s + g z_s \tag{1}$$

and

$$\mathrm{MSE}_{500}^* = c_p T_{500} + L_v q_{sat}(T_{500}) + g z_{500}, \tag{2}$$

respectively, whereby $c_p$ denotes the specific heat of air at constant pressure, $T_s$ and $T_{500}$ are the two-meter and 500 hPa temperatures, respectively, $L_v$ is the latent heat of vaporization, $q_s$ is the two meter specific humidity, $q_{sat}(T_{500})$ is the saturation

specific humidity of $T_{500}$, $g$ is the gravitational acceleration, and $z_s$ and $z_{500}$ are the height of the surface and the 500 hPa pressure level, respectively. Note that we approximate $q_{sat}(T_{500}) \approx \frac{\epsilon e_s(T_{500})}{500\,hPa}$, whereby $\epsilon$ is the molar ratio between water vapour and dry air and $e_s(T_{500})$ is the saturation vapour pressure at $T_{500}$ derived from the Clausius-Clapeyron relation.

As in Zhang and Boos (2023), convective instability is identified when $\mathrm{MSE}_{500}^* - \mathrm{MSE}_s \leq 0$. We examined $\mathrm{MSE}_{500}^* - \mathrm{MSE}_s$ at individual grid points as well as in a spatially aggregated manner and chose to present spatially aggregated $\mathrm{MSE}_{500}^* - \mathrm{MSE}_s$

below.

## 2.3   Computation of backward trajectories

The computation of backward trajectories in this study is exactly analogous to the approach of Röthlisberger and Papritz (2023a). We use the Lagrangian Analysis Tool LAGRANTO 2.0 (Sprenger and Wernli, 2015) to compute 15-day backward trajectories, started at each 3-hourly time step from each heatwave region (see Tab. 1 and text below) and at various vertical

levels. Trajectory information is stored at a 3-hourly temporal resolution as well. For the analysis of near-surface $T'$, trajectories are started at 10, 30, and 50 hPa above ground level in order to account for turbulent mixing and vertical variations in the PBL (e.g., Bieli et al., 2015). Furthermore, for the analysis of the vertical $T'$ structure, additional trajectories are started





between 1000 hPa and 175 hPa in 25 hPa steps above each heatwave region. No trajectories were started at grid points and
levels where the respective level intersected or was located below the local topography. Along each trajectory $(\boldsymbol{x}(t), t)$, where
$\boldsymbol{x}(t)$=(longitude($t$), latitude($t$), $p(t)$) at each trajectory time step $t$, the following variables are traced: $T$, $\overline{T}$, $\theta$, $q$, $\frac{\partial \overline{T}}{\partial t}$, and $\frac{\partial \overline{T}}{\partial p}$.
Thereby, the quantities $\frac{\partial \overline{T}}{\partial t}$ and $\frac{\partial \overline{T}}{\partial p}$ are computed using first-order finite differences.

## 2.4    Lagrangian $T'$ decomposition

The Lagrangian $T'$ decomposition of Röthlisberger and Papritz (2023a) builds conceptually on several previous studies that
evaluated the thermodynamic energy equation along kinematic backward trajectories to investigate how adiabatic warming
and diabatic heating affected the temperature in the air that subsequently contributed to near-surface temperature extremes
(e.g., Bieli et al., 2015; Santos et al., 2015; Quinting and Reeder, 2017; Zschenderlein et al., 2019; Papritz, 2020). The novel
aspect of the Röthlisberger and Papritz (2023a) approach is that it considers the material change of $T'$ instead of $T$ and thereby
also allows quantifying the effect of air parcel advection across horizontal gradients of $\overline{T}$ on $T'$. The decomposition of any
$T'(\boldsymbol{x}, t_f)$, where $t_f$ refers to the starting time of the trajectory, is obtained by re-writing the thermodynamic energy equation
in terms of $T'$ and then integrating it along a backward trajectory started at $(\boldsymbol{x}, t_f)$ from the time when $T'$ was last zero in the
respective air parcel (hereafter referred to as "genesis time", $t_g$) to $t_f$, i.e.,

$$T'(\boldsymbol{x}, t_f) = -\int_{t_g}^{t_f} \frac{\partial \overline{T}}{\partial t}\, d\tau - \int_{t_g}^{t_f} \boldsymbol{v} \cdot \boldsymbol{\nabla}_{\mathrm{h}} \overline{T}\, d\tau \quad + \int_{t_g}^{t_f} \left[\frac{\kappa T}{p} - \frac{\partial \overline{T}}{\partial p}\right] \omega\, d\tau + \int_{t_g}^{t_f} \left(\frac{p}{p_0}\right)^{\kappa} \frac{D\theta}{Dt}\, d\tau. \qquad (3)$$

Hereby, $\boldsymbol{v}$ is the horizontal wind, $\boldsymbol{\nabla}_{\mathrm{h}}$ the horizontal gradient operator, and $\kappa = \frac{R}{c_p}$ (see Röthlisberger and Papritz (2023a)
for a formal derivation of Eq. (3)). As in Röthlisberger and Papritz (2023a), the terms on the right-hand side of Eq. (3) are
hereafter referred to as seasonality $T'$, advective $T'$, adiabatic $T'$, and diabatic $T'$, respectively. For the details of computing
the individual terms in Eq. (3), the interested reader is referred to Röthlisberger and Papritz (2023a). Moreover, when evaluating
Eq. (3) for discrete trajectory data, a first residual appears because $T'$ is never exactly zero and thus $res1 = T'(\boldsymbol{x}(t_g), t_g)$[1].
A second residual $res2 = T'(\boldsymbol{x}, t) - \text{seasonality}\, T' - \text{advective}\, T' - \text{adiabatic}\, T' - \text{diabatic}\, T' - res1$ appears due to inaccuracies
in the computation of derivatives in Eq. (3). As in Röthlisberger and Papritz (2023a), we compute an overall residual $res =$
$res1 + res2 + \text{seasonality}\, T'$ in order to assess how well the $T'$ budget closes by just considering advective $T'$, adiabatic $T'$,
and diabatic $T'$. We find that $res$ is typically small compared to these three terms (for detailed information, see Röthlisberger
and Papritz, 2023a).

Furthermore, knowledge of the genesis time $t_g$ for any temperature anomaly $T'(\boldsymbol{x}, t_f)$ (i.e., the time when the respective
air parcel last had zero $T'$) allows computing the Lagrangian age of $T'(\boldsymbol{x}, t_f)$ as the difference between $t_f$ and $t_g$. Similarly,
the spatial scale, over which $T'(\boldsymbol{x}, t_f)$ formed, can be quantified by computing the Lagrangian formation distance as the great
circle distance between $\boldsymbol{x}(t_g)$ and $\boldsymbol{x}(t_f)$.

---

[1]As in Röthlisberger and Papritz (2023a), $t_g$ is defined as the last trajectory time step for which $T'(\boldsymbol{x}(t_g), t_g)$ has the same sign as $T'(\boldsymbol{x}(t_f), t_f)$, when
moving along the trajectory backwards in time.




**Table 1.** Definition of the case study regions during the PNW heatwave, the RU heatwave, and the UK heatwave.

|  | **Pacific Northwest (PNW)** 27 June–1 July 2021 | **Western Russia (RU)** 31 July–4 August 2010 | **Western Europe (UK)** 16 July–20 July 2022 |
|---|---|---|---|
| Latitude | 49° N–59° N | 48.5° N–58.5° N | 48.5° N–58.5° N |
| Longitude | 115° W–125° W | 41.5° E–51.5° E | 6° W–4° E |

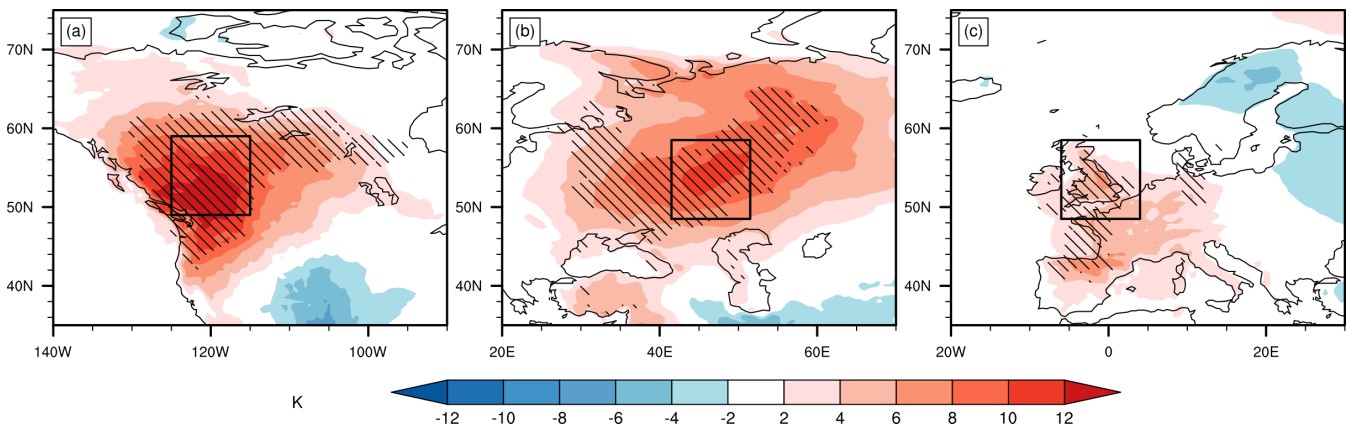

**Figure 1.** Temperature anomalies (5-day averages) on the second lowest model level during (a) the PNW, (b) the RU, and (c) the UK heatwaves (see Tab. 1 for the exact periods). Hatching in all panels denotes regions where the previous (since 1979) ERA5 two-meter temperature records were broken during the respective 5-day period. The black boxes define the regions we consider here for the three events.

## 2.5 Event definition

To define the temporal and spatial extent of the PNW, RU, and UK heatwaves, we applied the following procedure: First, we identify the grid point with the largest positive daily mean near-surface (second-lowest model level) temperature anomaly in the respective region and year with existing literature on heatwaves serving as a first approximation for the heatwave regions and periods (these dates and locations are hereafter referred to as peak locations and peak times). Then, we define for each event the temporal extent as the 5-day period centred on the respective peak day and the spatial extent as the 10.5° latitude by 10.5° longitude box centred on the respective peak location. This approach yields the heatwave periods and regions indicated in Tab. 1 and Fig. 1.

For the PNW heatwave, the selected region and period (27 June–1 July 2021) corresponds well with event definitions used in previous studies (e.g., Philip et al., 2022; Neal et al., 2022; Schumacher et al., 2022; Röthlisberger and Papritz, 2023a). Moreover, during the so-defined heatwave periods, surface temperature records were broken across the heatwave regions in all three cases (Fig. 1).





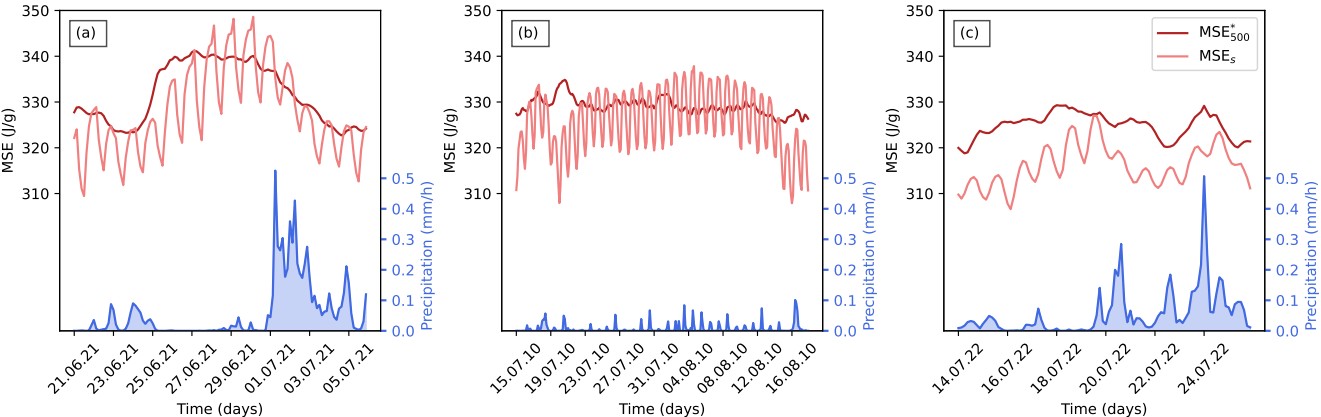

**Figure 2.** Surface moist static energy ($\mathrm{MSE}_s$, light red) and free tropospheric saturation moist static energy ($\mathrm{MSE}_{500}^*$, dark red) evolution during the (a) PNW heatwave, (b) RU heatwave, and (c) the UK heatwave. The blue line and shading depict precipitation (right y-axis). All values have been averaged over the respective $10.5°$ latitude by $10.5°$ longitude boxes indicated in Fig. 1.

We have extensively tested the sensitivity of our results to shifts in the spatial extent, location, and timing of the heatwave regions and periods. Qualitatively, our results are insensitive to horizontal shifts of the heatwave regions by a few degrees latitude and/or longitude, enlarging or shrinking the regions by a few degrees in either direction and shifts of the heatwave periods by 1–2 days, as long as the main synoptic events of interest are still contained within the heatwave regions and periods (not shown).

## 3 Results and discussion

### 3.1 Static stability during the PNW, RU, and UK heatwaves

We begin by examining to what extent the atmospheric vertical structure during the PNW, RU, and UK heatwaves was stable/unstable to moist convection (Fig. 2). Based on $\mathrm{MSE}_{500}^* - \mathrm{MSE}_s$, we find that, in particular during the PNW and RU heatwaves, the atmospheric vertical structure reached neutrality to moist convection (or was even convectively unstable) during their peak phases. Hereby, negative values of $\mathrm{MSE}_{500}^* - \mathrm{MSE}_s$ are somewhat surprising. However, at least during the PNW heatwave, they were conceivably related to particularly large convective inhibition (CIN) due to a strong boundary layer inversion (Neal et al., 2022, Fig. S1a). In addition, note that during the PNW heatwave, the near-surface temperatures peaked (i.e., stopped rising) when negative $\mathrm{MSE}_{500}^* - \mathrm{MSE}_s$ values started to appear, which is consistent with a convective limit to near-surface temperatures during this event. Furthermore, and also consistent with Zhang and Boos (2023), $\mathrm{MSE}_{500}^*$ peaked before $\mathrm{MSE}_s$ during the PNW heatwave and its termination was accompanied by significant precipitation, as well as a marked decrease in both $\mathrm{MSE}_{500}^*$ and $\mathrm{MSE}_s$. Similarly, during the UK heatwave $\mathrm{MSE}_{500}^*$ peaked on 18 July 2022, while $\mathrm{MSE}_s$ peaked on 20 July before the termination of the heatwave co-occurring with a peak in precipitation (Fig. 2c).



During the RU heatwave, the daily maximum of $\mathrm{MSE}_s$ regularly exceeded $\mathrm{MSE}^*_{500}$ during the afternoon time steps of its peak time, suggesting a convectively unstable stratification, which is also supported by (small amounts of) diurnal precipitation. Nevertheless, this moist convection did not terminate the RU heatwave. Thus, while convective instability may have limited the near-surface temperatures during the RU heatwave, Fig. 2b suggests that for the termination of a heatwave diurnal moist convection may not always suffice, in particular when moisture is strongly limited and soils have already dried out considerably, as was the case during the RU heatwave (Lau and Kim, 2012; Wehrli et al., 2019).

In summary, Fig. 2 suggests that convective instability, and with it the vertical temperature structure, played a key role in determining the magnitude and in part also the duration of the PNW, RU, and UK heatwaves. This motivates a detailed analysis of the characteristics and synoptic causes of the atmospheric vertical structure during these three events, which we pursue in the next sections. Specifically, we first discuss in detail the vertical temperature structure of the PNW heatwave and then compare and contrast it with that of the RU and UK heatwaves.

## 3.2 The PNW heatwave

### 3.2.1 Synoptic situation and evolution of near-surface $T'$

The synoptic evolution of the PNW heatwave has already been extensively examined in previous studies (e.g., Philip et al., 2022; Schumacher et al., 2022; Neal et al., 2022; White et al., 2023; Röthlisberger and Papritz, 2023a) and is thus only revisited here to a minimal extent to provide a synoptic context for the subsequent findings. A key synoptic ingredient to this event was an upstream cyclone which deepened rapidly between 22 and 25 June (visible in the top left of Fig. 3a). This cyclone produced a warm conveyor belt (WCB), which built up the extremely amplified (e.g., in terms of potential temperature on the dynamical tropopause, Oertel et al., 2023), upper-tropospheric quasi-stationary ridge, within which the heatwave occurred on the subsequent days.

In the early stage of the heatwave, between 25 and 26 June, near-surface $T'$ in the PNW heatwave region built up diabatically as well as advectively (Fig. 3i) and formed in air parcels that moved into the amplifying ridge and subsequently onshore (Fig. 3b). Since their respective $t_g$, most of these air parcels first ascended over the Pacific (whilst moving poleward into the ridge) and subsequently descended before they contributed to the positive near-surface $T'$ between 25 and 26 June (shown for selected trajectories in Fig. 3b).

Between 27 June and the peak of the heatwave on 30 June, the ridge-axis was located above the PNW heatwave region, where the average near-surface $T'$ exceeded +12 K (Fig. 3c, e). The better part of the $T'$ formed diabatically, but there was a gradual increase in the adiabatic contribution to near-surface $T'$ throughout the heatwave (Fig. 3i). This increase in the adiabatic $T'$ was also associated with a change in the behaviour of the trajectories of near-surface air, which increasingly started to spiral downward and anticyclonically near the PNW heatwave region (compare Figs. 3d and f).

After the peak of the heatwave (with spatially averaged near-surface temperature anomalies of roughly +15 K), on 30 June and 1 July, the total $T'$ as well as the diabatic $T'$ dropped significantly, consistent with the convective termination of the event and the associated onset of precipitation (see Fig. 2a). Note that even though the ridge started shifting eastward between 30





**Figure 3.** The synoptic evolution of the PNW heatwave daily averaged on (a, b) 25 June 2021, (c, d) 27 June 2021, (e, f) 29 June 2021, (g, h) 1 July 2021, and (i) the $T'$ and its contribution between 22 June and 3 July 2021. The left panels show the $T'$ at the second lowest model level in colours, the mean SLP in grey contours, the geopotential height at 500 hPa in purple contours, and the PNW heatwave region as a dashed black rectangle. The middle panels show 15 selected trajectories with positive $T'$ (coloured according to their pressure) arriving near the surface (each five at 10, 30, and 50 hPa above ground) on land grid points in the case study region at each date. Trajectories are only shown for trajectory times $t$ for which $t \geq t_g$. The right panel shows the near-surface $T'$ (black), advective $T'$ (green), adiabatic $T'$ (purple), and diabatic $T'$ (orange) averaged over land grid points of the PNW heatwave region from 24 June to 3 July 2021. The grey line denotes the maximum 5-day daily $T'$.





June and 1 July, it was only on 2 July that the advective $T'$ became negative, and thus indicates the advection of air from climatologically colder regions into the heatwave region (Fig. 3i). This indeed further underlines that the termination of the PNW heatwave was predominantly due to convective damping of near-surface $T'$ rather than due to changes in air mass advection into the region.

### 3.2.2 Time-mean vertical structure of $T'$

We first discuss time-mean characteristics of the vertical $T'$ structure of the PNW heatwave (Fig. 4). Previous studies already identified anomalous warmth throughout the troposphere during the PNW heatwave (Neal et al., 2022; Schumacher et al., 2022; Oertel et al., 2023; Zhang and Boos, 2023). Consistent with these results, we find positive 5-day average $T'$ in excess of +5 K throughout the troposphere, reaching up to 300 hPa (Fig. 4a). Applying the Lagrangian $T'$ decomposition to the entire volume of air above the PNW heatwave, from the surface to 200 hPa, allows quantitatively examining the physical causes and

vertical structure of these deep positive $T'$ (Fig. 4d–f). For $T'$ between 300 and 600 hPa, we find large positive contributions from advective $T'$ (exceeding +13 K) and diabatic $T'$ (roughly +9 to +13 K), as well as pronounced negative contributions from adiabatic $T'$ (less than -13 K, Fig. 4d-f). That is, after the respective anomaly genesis, these air parcels ascended in a poleward motion and thereby experienced considerable diabatic heating (presumably due to cloud formation), which is exactly the signature expected from a WCB. These results thus quantitatively substantiate the results of Neal et al. (2022), Schumacher

et al. (2022), and Oertel et al. (2023), who qualitatively argued that this positive temperature anomaly originated from the WCB.

The 5-day mean $T'$ peaked between 600 and 850 hPa with values larger than +13 K. In contrast to the air further aloft, these anomalies featured positive contributions from all three processes with roughly equal magnitude. That is, this air also moved from climatologically warmer regions towards the PNW heatwave region and experienced net diabatic heating since the

235 respective $t_g$ (albeit to a lesser extent than the air further aloft). However, contrary to the mid- to upper-tropospheric air, these air parcels experienced net subsidence since $t_g$, which manifested itself as positive adiabatic $T'$. This strong vertically aligned dipole, with positive and negative adiabatic $T'$ within an anticyclone, is perhaps surprising at first sight, as it is frequently argued that within anticyclones the subsidence contributes significantly to positive temperature anomalies in anticyclones (Pfahl and Wernli, 2012; Bieli et al., 2015; Zschenderlein et al., 2019). A close inspection of the trajectories' pressure evolution reveals

that, indeed, throughout the column above the PNW heatwave, the air subsided in the anticyclone (yielding a positive adiabatic $T'$ for the part of the trajectory since the lowest pressure was reached Fig. S2). However, what matters for the net effect of vertical motion on $T'$ (i.e., the total adiabatic $T'$) is the net vertical motion since the time when the air parcel last had zero $T'$, i.e., since $t_g$. For the PNW heatwave, we find that air parcels ending up in the heatwave region on levels above 600 hPa first experienced a strong and fast ascent within the WCB, which covered a larger pressure difference than the subsequent slow

descent within the anticyclone, yielding a net negative adiabatic $T'$. Conversely, for air parcels below 600 hPa, the subsidence within the anticyclone was larger than the ascent experienced on the way into the anticyclone, yielding positive adiabatic $T'$ at lower levels.



In the PBL, the contributions from adiabatic and, in particular, diabatic $T'$ jointly account for the bulk of the total $T'$. Hereby, this lower tropospheric diabatic $T'$ maximum is conceivably related to sensible and turbulent heat fluxes from the surface to the atmosphere, contrary to the upper-tropospheric diabatic $T'$ maximum, which is likely due to upstream cloud diabatic heating. Note that the large diabatic contribution to near-surface $T'$ found here is also in agreement with previous studies, who used an array of different methods to arrive at the same conclusion (Schumacher et al., 2022; Neal et al., 2022; Conrick and Mass, 2023).

By considering the Lagrangian age and formation distances (Fig. 4b, c), the $T'$ decomposition additionally allows quantifying the temporal and spatial scales over which these $T'$ form. For the mid- to upper-tropospheric air over the PNW heatwave region, we find ages of 8–10 days (Fig. 4b) and formation distances (Fig. 4c) of several thousand kilometres (at some levels exceeding 8000 km). These ages and formation distances are two to three times larger than those of near-surface $T'$.

### 3.2.3  Evolution of the vertical $T'$ structure

Next, we examine the specifics of the temporal evolution of the vertical $T'$ structure (Fig. 5). A first point to notice is that large positive $T'$ first emerged at upper levels and only several days later appeared near the surface. In accordance with Schumacher et al. (2022) and Zhang and Boos (2023), we find a first pulse of $T'$ formation at mid- to upper-tropospheric levels (exceeding 9 K) on 24 and 25 June, when in the PBL, $T'$ ranged between -5 K and +5 K (Fig. 5a). Positive advective and diabatic $T'$ and negative adiabatic $T'$ now quantitatively underline that this warming aloft was due to the aforementioned WCB, as qualitatively demonstrated by Neal et al. (2022), Zhang and Boos (2023), and Schumacher et al. (2022). This WCB signature between roughly 600 and 300 hPa persisted throughout the heatwave until 1 July.

Second, Fig. 5f shows evidence of diurnal heat accumulation in the PBL, as observed in earlier mega heatwaves by Miralles et al. (2014). The PBL height exhibits a pronounced diurnal cycle, while the vertical extent of the lower tropospheric diabatic $T'$ maximum remains almost constant. That is, at night, the positive diabatic $T'$ from the PBL evidently remained in the residual layer. We thus hypothesise that on the following day, when the PBL grew again in depth, the air from the residual layer was again mixed into the PBL, which conceivably resulted in multi-day heat accumulation as described by Miralles et al. (2014).

Third, there is clear evidence of downward propagation of positive adiabatic $T'$ in the lower troposphere (Fig. 5e). Interestingly, the peak in near-surface $T'$ (around 30 June 2021) coincided with the time when the peak adiabatic $T'$ reached the PBL. This suggests that some of the air that was mixed into the PBL during the PBL growth had been significantly heated adiabatically before.

Fourth, the age of mid- to upper-tropospheric $T'$ as well as PBL $T'$ increased considerably over the course of the PNW heatwave (Fig. 5b). On 25 June, the age of $T'$ at 400 hPa was 4–6 days and 2–4 days at 900 hPa. By 30 June, Lagrangian ages of 400 hPa $T'$ reached 10–12 days, whilst at 900 hPa $T'$ was 6–8 days old. Therefore, ageing of $T'$ is observable throughout the tropospheric column above the PNW heatwave.

In summary, the temporal evolution of the vertical $T'$ structure supports a top-down control of near-surface $T'$ via convective (in)stability, quantitatively demonstrates that the peak of the heatwave occurred after a multi-day period of diurnal heat accumulation (as described in Miralles et al., 2014) and coincided with the time when strongly adiabatically warmed air was





**Figure 4.** Five-day mean (27 June–1 July 2021) vertical cross-section showing the $T'$ decomposition during the PNW heatwave. Fields are latitudinally averaged between 49–59° N over land grid points. The panels show (a) $T'$, (b) the Lagrangian age, (c) the formation distance, (d) the advective $T'$, (e) the adiabatic $T'$, and (f) the diabatic $T'$ in colour. The black line depicts the dynamical tropopause (2 PVU contour; $1\,\mathrm{PVU} = 10^6\,\mathrm{m}^2\,\mathrm{s}^{-1}\,\mathrm{K}\,\mathrm{kg}^{-1}$), grey contours show potential temperature, and the purple line shows the PBL height. In addition, the topography is shown in grey. The two black vertical lines indicate the longitudinal extent of the PNW heatwave region.

mixed into the PBL (Schumacher et al., 2022). Furthermore, Fig. 5b, for the first time, documents the ageing of $T'$, which has been qualitatively surmised by previous studies putting forward the concept of a "heat dome" (Neal et al., 2022; Zhang et al.,



**Figure 5.** Time-height plots showing the $T'$ decomposition during the PNW heatwave spatially averaged over land grid points in the PNW heatwave region between 22 June and 3 July 2021. The panels show (a) $T'$, (b) the Lagrangian age, (c) the formation distance, (d) the advective $T'$, (e) the adiabatic $T'$, and (f) the diabatic $T'$ in colour. The black line shows the dynamical tropopause (2 PVU contour), the grey contours depict the potential temperature, and the purple line indicates the PBL height.

2023). In the next sections, we contrast the characteristics and evolution of the vertical $T'$ structure during the PNW heatwave
with that of the RU and UK heatwaves.





**Figure 6.** As 3 but for the RU heatwave. Panels are shown for (a, b) 1 August 2010, (c, d) 3 August 2010, (e, f) 5 August 2010, (g, h) 7 August 2010, and (i) 28 July–8 August 2010.

## 3.3 Comparing the PNW, RU, and UK heatwaves with regard to their vertical $T'$ structure

### 3.3.1 Comparing and contrasting the RU and PNW heatwaves

Another record-shattering and intensely studied heatwave occurred in western Russia in 2010, lasting from mid-July to mid-August (Barriopedro et al., 2011; Dole et al., 2011; Lau and Kim, 2012). This heatwave was associated with an exceptionally

long-lasting and stationary anticyclone (first column of Fig. 6, Barriopedro et al., 2011), which was embedded within a larger-



scale Rossby wave train (Lau and Kim, 2012; Trenberth and Fasullo, 2012; Schneidereit et al., 2012). Here, we briefly describe the formation of near-surface $T'$ as well as the vertical $T'$ structure during the peak time of the RU heatwave and then contrast these results to those discussed before for the PNW heatwave.

The trajectories of air parcels contributing to the near-surface RU heatwave between 31 July and 4 August 2010 subsided in an anticyclonically spiralling motion prior to reaching near-surface levels (Fig. 6b, d, f, h). During the 5-day maximum $T'$ (grey coloured bar in Fig. 6i), the diabatic $T'$ exhibited a pronounced diurnal cycle and, in terms of daily means decreased slightly, from +7 K to +5 K, while the adiabatic $T'$ increased from +3 K to +7 K. At the same time, the advective $T'$ remained near zero. Throughout the 5-day period, the adiabatic and diabatic contributions were of comparable magnitude. While these results are consistent with those of Zschenderlein et al. (2019), they are in some contrast to other studies that emphasised, in particular, the importance of diabatic processes due to prolonged soil moisture depletion (e.g., Trenberth and Fasullo, 2012; Barriopedro et al., 2011; Miralles et al., 2014; Hauser et al., 2016). This is particularly noteworthy, as, both in absolute and relative terms, the RU heatwave featured far smaller diabatic contributions to its near-surface $T'$ than the PNW heatwave (compare Figs. 6i and 3i). Moreover, the adiabatic contribution grew steadily throughout the evolution of the heatwave and eventually clearly exceeded the diabatic contribution.

The vertical $T'$ structure of the RU heatwave featured a number of similarities to that of the PNW heatwave (compare Figs. 7 and 4). For instance, during both heatwaves, the positive $T'$ extended throughout the troposphere (see also Fig. S2 in Zhang and Boos, 2023). Moreover, during both heatwaves mid- to upper-tropospheric $T'$ featured large positive advective contributions (Figs. 4d and 7d), while large positive adiabatic $T'$ was restricted to the lower troposphere. Furthermore, a clear maximum in diabatic $T'$ occurred near the surface (Fig. 7f).

However, also clear differences emerge that point to differing synoptic dynamics involved in the two events: The prominent WCB signature aloft (positive advective and diabatic $T'$, negative adiabatic $T'$) during the PNW heatwave was not apparent during the RU heatwave. Rather, the diabatic $T'$ in the upper troposphere during the RU heatwave was negative, presumably due to radiative cooling. These results suggest that the anticyclone associated with the RU heatwave was not as strongly diabatically driven as the one during the PNW heatwave. Moreover, note that the almost isobaric flow in the upper troposphere during the RU heatwave (not shown) is consistent with barotropic Rossby wave dynamics as the main driver of this anticyclone.

The two heatwaves also differed considerably regarding the Lagrangian age and formation distances of their $T'$. During the PNW heatwave, the oldest anomalies (>8 days) were located between 600 and 300 hPa, whereby those anomalies featured the WCB signature. During the RU heatwave, the oldest $T'$ (>10 days) were located below 600 hPa and, in contrast to the oldest anomalies during the PNW heatwave, featured comparatively small formation distances of less than 2000 km (Fig. 4c). That is, in contrast to the oldest anomalies in the PNW heatwave, these oldest anomalies of the RU heatwave built up quasi-locally, whilst recirculating and subsiding in the anticyclone. In summary, these contrasts regarding the WCB signature in the $T'$ composition of mid- to upper-level $T'$ during the PNW and RU heatwaves show that an upstream WCB may amplify an anticyclone within which subsequently a major heatwave occurs (e.g., during the PNW heatwave), but this is clearly not a necessary condition for a major heatwave to occur (e.g., the RU heatwave).





**Figure 7.** As Fig. 4 but for the RU heatwave. Latitudinal averages are taken over 48.5–58.5° N over the period from 31 July to 4 August 2010.

Next, we contrast the evolution of the vertical $T'$ structure during the RU and PNW heatwaves (compare Figs. 5 and 8). During the RU heatwave, positive $T'$ extended throughout almost the entire troposphere with far less temporal variation than during the PNW heatwave, which is unsurprising given that the RU heatwave lasted for over a month (Barriopedro et al., 2011;





**Figure 8.** As Fig. 5 but for the RU heatwave between 28 July and 8 August 2010. Spatial averages are taken over land grid points in the RU heatwave region.

Dole et al., 2011; Lau and Kim, 2012; Zhang and Boos, 2023). Hereafter, we focus on the selected peak period from 31 July to 4 August 2010.

By and large, the advective, adiabatic, and diabatic $T'$ featured only small temporal variations in contrast to the PNW heatwave, particularly in the free troposphere. There, advective $T'$ was positive above 500 hPa, with small values below,



adiabatic $T'$ was positive below 500 hPa, with small values above and diabatic $T'$ was consistently negative throughout the entire free troposphere. The unusual persistence of the RU heatwave was thus also reflected in a rather stationary and persistent vertical $T'$ structure.

However, similarly to the PNW heatwave, the PBL height during the RU heatwave featured a pronounced diurnal cycle and regularly reached nearly 700 hPa during the afternoon. As also documented by Miralles et al. (2014), evidence for diurnal heat accumulation (i.e., positive diabatic $T'$ persisting overnight in a residual layer above the PBL) is apparent predominantly between 28 and 31 July 2010, i.e., when near-surface $T'$ was still accumulating. Later during the heatwave (in early August), large positive adiabatic $T'$ but small diabatic $T'$ above the nighttime PBL suggest that the maintenance of extremely large PBL

$T'$ was aided by the mixing of adiabatically heated air into the diurnally growing PBL. However, a key difference between the PNW and RU heatwaves in the PBL concerns the depth of the vertical layer with positive diabatic $T'$ near the surface, which was far shallower during the RU compared to the PNW heatwave. It is currently unclear how exactly this result is to be interpreted.

Surprisingly, the diabatic $T'$ declined within the lowermost 200 hPa from 29 July onwards, while the adiabatic contribu-

tions grew steadily. This contrasts the findings of Miralles et al. (2014) and Zschenderlein et al. (2019), who emphasised the importance of sensible heat fluxes during the RU heatwave, and also distinguishes it from the PNW heatwave.

Finally, similarly to the PNW heatwave also during the RU heatwave, $T'$ in the lower troposphere were clearly ageing (Fig. 8b) throughout the course of the event. For instance, at 950 hPa, the age of $T'$ increased from 4–6 days on 30 July to more than 10 days on 8 August.

### 3.3.2   Contrasting the UK heatwave to the RU and PNW heatwaves

As a third case study, we examine the western European and UK heatwave in July 2022, during which at 46 stations in the UK, the previous nation-wide temperature record of 38.7° C was broken (Press Office, 2022). Before the heatwave hit the case study region, positive near-surface $T'$ formed over the Iberian Peninsula under a subtropical ridge (Fig. 9a) and downstream of a cutoff cyclone located in the eastern North Atlantic. The ridge then further extended to the north and moved eastwards

(Fig. 9c, e, g) towards the UK and western Europe. Throughout the event, large positive near-surface $T'$ were first found on the Iberian Peninsula, then over the British Isles, followed by Germany (Fig. 9).

During the 5-day $T'$ maximum in our selected heatwave region, the adiabatic $T'$ contribution was dominant (Fig. 9i), which agrees with the findings of Bieli et al. (2015) and Zschenderlein et al. (2019), who studied air mass origins during heatwaves over the British Isles climatologically and emphasised the importance of subsidence for these events. However, the advective $T'$

contribution, exceeding +4 K on 19 July, was also substantial. Furthermore, in stark contrast to the other two events, the diabatic $T'$ is near-zero or even negative (Fig. 9i). One of the reasons for this discrepancy might be that during the UK heatwave, near-surface air parcels approached the UK from the North Atlantic and on near-surface levels (Fig. 9b, d, f, h), where anomalously warm air is typically cooled through sensible heat fluxes (Röthlisberger and Papritz, 2023a). Finally, note that the UK heatwave lasted considerably shorter than the other two heatwaves and ended swiftly after the 5-day $T'$ maximum.





**Figure 9.** As 3 but for the UK heatwave. Panels shown for (a, b) 15 July 2022, (c, d) 17 July 2022, (e, f) 19 July 2022, (g, h) 21 July 2022, and (i) 14 July–22 July 2022. The positive $T'$ trajectories in panels (b, h) are particularly short since positive $T'$ are only a few hours old.

Next, we examine the time-mean vertical $T'$ structure during the UK heatwave (Fig. 10). In the 5-day mean, positive $T'$ extended throughout the troposphere, similarly to the PNW and RU heatwaves (Fig. 10a). The maximum $T'$ was located around 875 hPa. Above the PBL, the UK heatwave featured a qualitatively similar vertical $T'$ structure as the RU heatwave, with a layer of positive adiabatic $T'$ extending from the surface to roughly 700 hPa (500 hPa during the RU heatwave, Figs. 7e and 10e, note that such a layer with positive adiabatic $T'$ was also apparent during the PNW heatwave, Fig. 4e), and above that

positive $T'$ consisting of positive advective $T'$ and modest negative contributions from adiabatic and diabatic $T'$.



**Figure 10.** As Fig. 4 but for the UK heatwave. Latitudinal averages are taken over 48.5° N–58.5° N over the peak period from 16 July to 20 July 2022.

A clear contrast between the UK heatwave and the other two events is that the diabatic $T'$ was negative in the heatwave region, even near the surface, and ranged between near 0 K and -5 K in the free troposphere (Fig. 10f). Thereby, the near-surface diabatic $T'$ increased towards the east, i.e., over European land regions (Fig. 10e). Furthermore, the age of $T'$ during the UK heatwave was considerably smaller than during the two other events, in particular for the largest $T'$ occurring in the





lowermost 300 hPa (less than 6 days during the UK heatwave compared to more than 6 and 8 days during the PNW and RU heatwaves, respectively, Figs. 4b, 7b, and 10b). Note, however, that in both the UK and RU heatwaves, the oldest $T'$ were found in the lower troposphere, while the largest formation distances occurred at upper levels (Figs. 7b, c and 10b, c). In summary, the small adiabatic and diabatic $T'$ at upper levels, as well as corresponding modest ages and formation distances during the peak period of the UK heatwave indicate that its upper-level $T'$ was not significantly affected by a WCB.

Finally, we turn our attention to the temporal evolution of the vertical $T'$ structure during the UK heatwave (Fig. 11). Interestingly, it bears some resemblance to that of the PNW heatwave (Fig. 5) and does feature some WCB characteristics preceding the peak period. Around 15 July, i.e., 1–2 days before the positive near-surface $T'$ started to form, positive $T'$ developed between 500 and 400 hPa. Figure 11 reveals that these $T'$ consisted of positive diabatic $T'$ of +5 to +9 K (Fig. 11f), negative adiabatic $T'$ (Fig. 11e), and positive advective $T'$ (Fig. 11d), as well as comparatively long formation distances and

large ages (Fig. 11b, c), which is reminiscent of the WCB signature during the PNW heatwave (Fig. 5). Much like during the PNW heatwave, upstream cloud diabatic heating and the associated WCB-like air stream thus helped building up the ridge within which the UK heatwave ultimately occurred. Such WCB-induced ridge amplification is a common feature of heatwaves in this region (Zschenderlein et al., 2020). Moreover, the formation of positive $T'$ first at upper levels is consistent with an upper-level control on near-surface $T'$ (Zhang and Boos, 2023), although, as mentioned above, significant near-surface

diabatic $T'$ was lacking in this case. From 16 July onwards, i.e., during the peak phase of the heatwave, the positive $T'$ above 700 hPa formed exclusively through advection. During this stage, the vertical $T'$ structure in the mid- to upper-troposphere thus resembled that of the RU heatwave more than that of the PNW heatwave.

    The $T'$ peaked on 19 July at levels below 850 hPa (Fig. 11a). Interestingly, as for the PNW heatwave, the peak in adiabatic $T'$ propagated downward, and the timing of the peak of near-surface $T'$ coincided with the arrival of large positive adiabatic $T'$ at

near-surface levels (Fig. 11e). Similar to the other two events, the $T'$ in the lowermost 300 hPa aged by about 2-4 days from 16 July to 19 July (Fig. 11b). Hereby, in the cross-sections shown in Fig 11 the ageing of $T'$ (below 700 hPa) was co-located with the downward propagating and amplifying peak in adiabatic $T'$. Finally, note that the ageing of $T'$ was confined predominantly to near-surface levels during both the RU and UK heatwaves, which contrasts the PNW heatwave, where the ageing of $T'$ also occurred between 600 and 300 hPa.

The key difference between the UK heatwave and the other two events is, however, that at all levels, the diabatic $T'$ was small and mostly negative, except for the WCB-like signal between 500 and 300 hPa that preceded the near-surface heatwave. We interpret the lack of positive near-surface diabatic $T'$ as a result of the proximity of the UK heatwave region to the North Atlantic Ocean. Note that even climatologically, the diabatic $T'$ is near zero during heat extremes in the UK and adjacent European coastal regions (Röthlisberger and Papritz, 2023a). This is because the respective (already anomalously warm) air

is transported over ocean surfaces and is thereby cooled diabatically before arrival in these regions. That is, near-surface $T'$ during heat extremes that consist predominantly of adiabatic $T'$ and some advective $T'$ appear to be the norm for the UK and adjacent European coastal regions.





**Figure 11.** As Fig. 5 but for the UK heatwave between 14 July and 21 July 2022. Spatial averages are taken over land grid points in the 10.5° latitude by 10.5° longitude box indicated in Fig. 1c.

## 4  Synthesis and conclusions

Motivated by recent studies arguing for a top-down control on maximum near-surface temperatures during major heatwaves

(Neal et al., 2022; Zhang and Boos, 2023), we have examined the detailed vertical temperature structure of three recent, record-shattering heatwaves that occurred in the Pacific Northwest in 2021 (PNW), western Russia in 2010 (RU), and western Europe





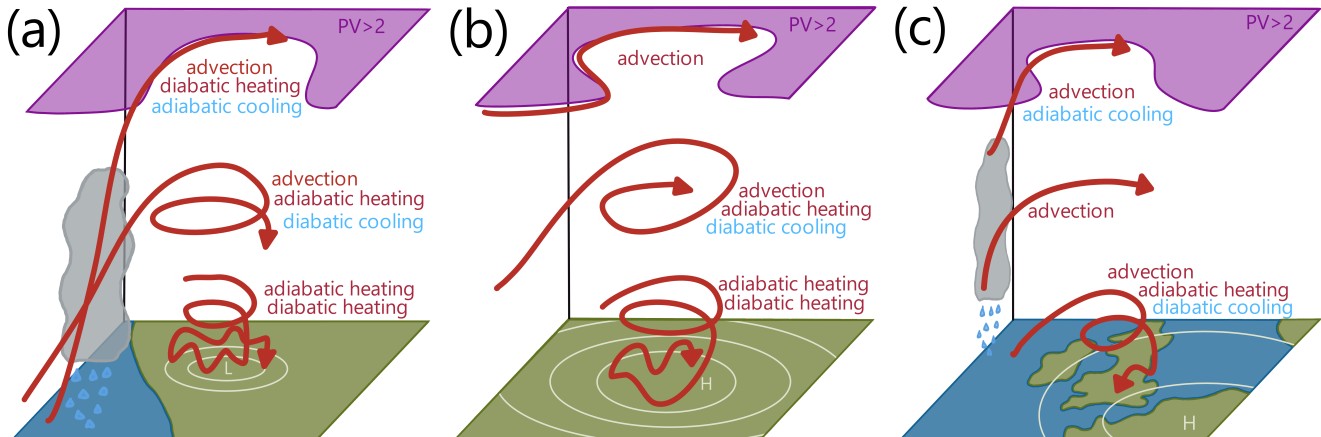

**Figure 12.** Schematic of the (a) PNW, (b) RU, and (c) UK heatwaves. The three red lines in each of the panels depict the characteristics of air parcels ending up in the upper and middle troposphere, as well as in the boundary layer (the wriggling denotes the diurnal cycle). Processes acting to increase and decrease the ultimately positive $T'$ are denoted in red and blue font colour, respectively. The purple shading denotes stratospheric air.

in 2022 (UK). Hereby, a top-down control on the maximum near-surface $T$ (and $T'$) through moist-convective (in)stability is clearly evident for the PNW and RU heatwaves, consistent with the theoretical arguments by Zhang and Boos (2023). This top-down control was less pronounced in the case of the UK heatwave, presumably due to weaker near-surface diabatic

heating such that the moist-convective limit was never closely approached. To understand the physical causes of the vertical $T'$ structure during these three events, we employed the diagnostic framework of Röthlisberger and Papritz (2023a) and quantified the contributions to $T'$ of advection of air across climatological temperature gradients, adiabatic warming, and diabatic heating.

Our analyses reveal that all three heatwaves were characterized by bottom-heavy yet vertically deep positive $T'$ extending from the surface across the entire tropospheric column. The intensity of $T'$ and the underlying driving processes vary to a

large degree in the vertical as well as from one case to the other (summarized in Fig. 12). Hereby, the differences across the three heatwaves relate to different synoptic storylines. The PNW heatwave was affected heavily by an upstream cyclone and its associated warm conveyor belt (WCB), which was instrumental for the amplification and poleward extension of the quasi-stationary ridge and also contributed significantly to upper-level $T'$ (Fig. 12a). In contrast, the anticyclone within which the RU heatwave formed was primarily affected by large-scale (dry) Rossby wave dynamics and experienced little upstream diabatic

heating (Fig. 12b). The UK heatwave can perhaps be considered as an intermediate case, where upstream cloud diabatic heating played some role in the early amplification of the anticyclone but was presumably not as dominant as in the PNW case.

Regarding the vertical structure of the $T'$ decomposition during the three heatwaves, our principal findings are as follows:

– In all three cases, the composition of mid- and upper-tropospheric positive $T'$ differs substantially from that of lower tropospheric positive $T'$. Mid- and upper-tropospheric positive $T'$ forms due to a combination of positive advective $T'$,

resulting from the poleward advection of air from climatologically warmer regions and a combination of either near zero




adiabatic $T'$ and negative diabatic $T'$ or strongly negative adiabatic $T'$ and positive diabatic $T'$. The latter corresponds to the characteristic signature of upstream, strongly ascending WCB air streams, which not only contribute to ridge amplification but are also a source of substantial $T'$ aloft. This WCB signature was particularly pronounced for positive mid- to upper-tropospheric $T'$ during the PNW heatwave, and our results thus now quantitatively support earlier studies that qualitatively argued for the importance of this WCB (Neal et al., 2022; Schumacher et al., 2022; Oertel et al., 2023). Moreover, a WCB also contributed to ridge amplification and positive upper-level $T'$ during the early phase of the UK heatwave, whereas in the case of the RU heatwave, such a signature was entirely absent. Adiabatic warming contributed notably to lower tropospheric $T'$ during all three heatwaves, acting in conjunction with diabatically generated $T'$ in the case of the PNW and RU heatwaves.

– Our results provide a nuanced view regarding the effect of vertical motion on $T'$ in anticyclones. While previous studies have argued that subsidence within anticyclones leads to positive $T'$ without specifying the vertical level (Pfahl and Wernli, 2012; Zschenderlein et al., 2019), we show that the net effect of vertical motion on $T'$ in the anticyclones leading to the three heatwaves is positive only in the lower 200–300 hPa of the atmosphere. Above that layer, the air does subside within the upper-level ridge. However, since a non-negative $T'$ starts to form already earlier while the air ascends – either quasi-isentropically or cross-isentropically if cloud formation occurs – a net negative effect of vertical motion on $T'$ results.

– Amplification and downward propagation of adiabatic $T'$ was evident in both the PNW and UK heatwaves, and near-surface $T'$ peaked when the maximum of adiabatic $T'$ reached near-surface levels. Subsequent studies should thus analyse a large number of major heatwaves to assess to what extent the timing of heatwave peaks is determined by the arrival of air with the largest positive adiabatic $T'$ in the boundary layer (as also observed for the PNW heatwave by Schumacher et al., 2022).

– Finally, for all three cases, we find evidence of ageing of $T'$, in particular for lower-tropospheric air that subsided significantly before contributing to its respective heatwave. Thus, qualitatively this supports the notion of a "heat dome" in which air re-circulates and accumulates heat.

The results of our study are limited in a number of ways. First, the total $T'$ is often the residual of partly opposing advective, adiabatic, and diabatic contributions. In those cases, it is difficult to determine which process should be considered as the main cause of the total $T'$, as the three processes are generally anti-correlated through their physical linkages (Röthlisberger and Papritz, 2023b). For cold extremes Röthlisberger and Papritz (2023b) found that the most intense events, with the largest negative near-surface $T'$, occurred when these cancellations were particularly weak, i.e., when there was little damping of negative near-surface $T'$. A similar reasoning might also apply to near-surface $T'$ during hot extremes, and subsequent studies should thus examine more generally to what extent anomalously strong "forcing" (e.g., advection or dynamically induced subsidence) causes unusually large positive $T'$ and to what extent hot extremes result from attenuated (convective or other) damping mechanisms. Second, even though we use ERA5 data at a relatively high spatial and temporal resolution of $0.5°\times0.5°$ and 3 hours,



respectively, we cannot entirely assess to which extent the results of our trajectory calculations are resolution-dependent. This
particularly concerns all results and arguments related to boundary layer dynamics and convection, which clearly cannot fully
be resolved at this resolution. However, while this is a caveat of our analysis, note that the spatial and temporal resolution used
here is at least as high as that used in previous trajectory-based studies on heat extremes (e.g., Bieli et al., 2015; Quinting and
Reeder, 2017; Zschenderlein et al., 2019; Schumacher et al., 2022). Third, our analyses focus almost entirely on $T'$, while for
reaching moist convective neutrality also the vertical humidity structure is of obvious importance, which, according to Zhang
and Boos (2023) thus, also plays a role in limiting near-surface temperatures during major heatwaves. We fully acknowledge
that subsequent studies should also assess the vertical moisture structure during major heatwaves and examine its causes.

Despite these caveats and a clear need for subsequent, more systematic studies on the atmospheric vertical structure during
major heatwaves, our quantitative analysis of $T'$ profiles and their physical causes already yields considerable novel insights
into the four-dimensional functioning of anticyclone-heatwave couples. It is thus hoped that these analyses will serve as a
starting point for subsequent studies examining more systematically how large-scale dynamics, boundary layer processes as
well as convection act in concert to generate the most intense and most impactful heatwaves.

*Code and data availability.* All results are based on the ERA5 reanalysis from ECMWF, which can be downloaded from the Copernicus
Climate Service (Copernicus Climate Service, 2021, https://climate.copernicus.eu/climate-reanalysis). The LAGRANTO 2.0 code is freely
available from Sprenger and Wernli (2015). The temperature anomaly decomposition was calculated by using code of Röthlisberger and
Papritz (2023a, https://doi.org/10.3929/ethz-b-000571107). Python scripts used to produce the analysis and visualisations are available from
the authors upon request.

*Author contributions.* BH performed the analyses and drafted the figures. MR and LP conceived the study. All three authors jointly inter-
preted the results and jointly wrote the paper.

*Competing interests.* LP is a member of the editorial board of *Weather and Climate Dynamics*. The peer-review process was guided by an
independent editor, and the authors also have no other competing interests to declare.

*Acknowledgements.* We thank the ECMWF for providing access to the ERA5 reanalyses. MR acknowledges funding from the European
Research Council under the Horizon 2020 research and innovation program (project INTEXseas, grant no. 787652).



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
