# Peer review of "Understanding the vertical temperature structure of recent record-shattering heatwaves"

_EGUsphere, 2023_

## Referee Comment (RC1)

**Review of "Understanding the vertical temperature structure of recent record-shattering heatwaves" by B. Hotz, L. Papritz and M. Röthlisberger submitted to Weather and Climate Dynamics**

**General comment:**

In this paper, the authors analyse the vertical temperature structure of three record-breaking heatwaves. In a first step, they use the diagnostics developed by Zhang and Boos (2023) to asses to what extent convective stability/instability played a role in determining the magnitude and duration of the heatwaves. In a second step, the authors then perform a detailed Lagrangian analysis based on the diagnostics developed in Röthlisberger and Papritz (2023). They quantify to what extend horizontal advection, vertical advection, and diabatic heating contributed to the specific events. By doing so, they go beyond many other Lagrangian-based studies in that they look at the entire vertical structure and not just the near ground. They find that the contribution from the individual processes varies significantly across the troposphere, with horizontal advection generally being a key process for establishing positive temperature anomalies in the mid to upper troposphere and adiabatic and diabatic warming in the lower troposphere; whereby large differences between the events have also occurred. Many of the results are consistent with the existing literature; in some cases the authors find contradictions. For sure, the analysis will stimulate future work in understanding the formation mechanisms of heatwaves.

The manuscript is well-thought, well-written, and really worth reading. I especially enjoyed those parts that dealt with the Lagrangian analysis, which corresponds to the better part of the manuscript. Generally, I'm a fan of the Röthlisberger-Papritz-diagnostics and I'm convinced that its application brings us a good deal closer to understanding the underlying mechanisms in the development of temperature anomalies. Unfortunately, I had difficulties to follow the reasoning in Section 3.1, which deals with the role of convective instability, and how it relates to the (results of the) Lagrangian analysis.

Below I have compiled a list of questions and comments, and I am eager to hear the authors' responses. If properly revised, I find the manuscript well suited for publication.

**Major comments:**

Section 3.1: I have some difficulty following your reasoning in this section. My main problem is that I thought Zhang and Boos (2023) were arguing that a heat wave persists as long as the atmosphere is stable for moist convection, and that once convective instability is reached, precipitation sets in, ending the heat wave. However, for the PNW and the RU heatwaves you do see convective instability, but this does not end the heatwaves. It might help my understanding if you could describe what the plots in Figure 2 should look like to conclude that convective instability did not play a significant role in the heatwaves. Also, addressing my questions about L173-175 and L176/176 will certainly be helpful as well.

L173-175: Here you write "Based on MSE*500-MSEs we find that during the PNW and RU heatwaves, the atmospheric vertical structure reached neutrality to moist convection during their peek phases. Hereby, negative values of MSE*500-MSEs are somewhat surprising." But I thought you were inferring from the negative values of the MSE*500 MSEs that the vertical structure achieved neutrality. So why are the negative values surprising if the atmospheric vertical structure has reached neutrality? My understanding is that it would be better if you wrote, e.g., "During the PNW and RU heatwaves, we find negative values of MSE*500 MSEs during their peak phases, indicating that the atmospheric vertical structure has achieved neutrality to moist convection at that time. This is somewhat surprising."

L176/176: You write that one should "note that […] the near-surface temperatures peaked when negative […] values started to appear". But I cannot see any line in Fig. 2 indicating temperature. I think this is also the reason why it's pretty hard for me to follow your reasoning in the whole section.

L266-270: Here, it's not clear to me why the pronounced diurnal cycle of the PBL height in combination with the constant vertical extent of the lower tropospheric diabatic T' shows that there is heat accumulation in the PBL. Could you explain this a bit more? E. g. What would the PBL height and diabatic T' look like if there had been no heat accumulation?

L279-281: As mentioned earlier, I'm not entirely convinced that you see evidence for a top-down control of T' via convective stability/instability, nor for the multi-day heat accumulation. But this probably changes if you clarify this a bit more in the earlier paragraphs.

**Minor comments:**

Title: My feeling is that "understanding" is a very strong word that means to know the causes/reasons why things are the way they are. With your diagnostics, you can attribute a certain process to a certain T' and this is really great. But, is it enough to really understand the vertical temperature structure, i. e. to give a cause for it? In light of that (and your comment in L455458), you might consider rephrasing the title a bit, e.g. into "Towards an understanding of the vertical temperature structure of ...". Or: "Disentangling the vertical temperature structure of ...".

L23/24: I think what you do is that you analyse the large-scale dynamics (advection, subsidence, WCB), the moist convection (stability/instability), and a bit the boundary layer processes (PBL height), but at the moment I do not see that you really discuss their "interplay". If you think so too, consider rephrasing this sentence.

L51: Which events do you mean with "events mentioned above"? The PNW, RU, and UK heatwaves or the "benign" heat waves from the previous paragraph?

L91/125: Here, I'm interested in a technical detail. If you compute the mean T on modellevels, what do you then take as $\Delta p$ in the computation of the vertical gradient? E. g. do you compute a climatological mean pressure for each modellevel at each grid point, from which you then infer the $\Delta p$ between each modellevel at each grid point? Or do you take the instantaneous values of p?

L93: I do not fully understand what you mean by "9-year windows centered on the time step of interest". Especially I was wondering how you handle dates at the edges of the ERA5 timeseries when there are no 4 years left, e. g., dates in the year 2021 or 2022. Could you clarify this?

L121: Can the LAGRANTO trajectories account for turbulent mixing in the PBL?

L125: I was just wondering why you trace specific humidity q. I suppose just in case it's needed at some point? If so, you could drop q in this listing here to prevent confusion.

L160/161: Do the regions and periods of the other two heatwaves also correspond to the event definitions in other studies?

L166/167: This goes back to my previous comment. You write that your results are "insensitive to horizontal shifts of the heatwave regions by a few degrees, enlarging or shrinking the regions by a few degrees, and shifts of the heatwave periods by 1-2 days." For the PNW and the UK heatwaves, I can well imagine that this ist true. However, for the RU heatwave, which was quite extensive spatially and lasted over a month in total, your results may not be representative of the entire heat wave period/region. So it could well be that a shift of a few days doesn't matter, but a shift of, say, 10 days might actually matter and your results would have looked different if, e. g., you had chosen a period closer to the onset phase of the heatwave. I think you should mention this a bit more prominently (see also my comment to L347).

L206: In Fig. 3b, I can't see that the air parcels first ascended and then descended. I think this is because the trajectories overlap and cover each other and only a small part of the colorbar is used. I think the figure would improve if you adjusted the colorbar accordingly and made the trajectories a little transparent. Maybe it would also be a good idea to take trajectories that end up more apart. The same suggestions apply to Fig. 6 and 9.

L254: Determining the Lagrangian age and formation distances would be possible without doing the full T' decomposition. Thus, reformulating this sentence into e.g. "By determining the Lagrangian age and formation distances, the temporal and spatial scales over which the temperature anomalies form can be quantified." might be more precise.

L283 (and L21 and L453): Here you argue that the aging of T' suggests the concept of a "heat dome" in which "air recirculates and accumulates heat". I agree with you that a heat dome would have exactly these properties, i.e., aging of T', and that it is plausible to observe this during the PNW heatwave. However, the

aging of T' is not conclusive evidence that the air masses are actually recirculating. What can "only" be seen is that the air masses that happen to arrive in the heat wave region tend to be older (in terms of temperature anomaly lifetime) than before. But you can't be sure if it's always the same air masses you observe, or if it's new air masses that happen to have an older lifetime. If you agree, I would appreciate a comment on this when you make statements about the "heat dome".

Fig. 5/8/11c: Never mentioned/discussed in the text.

L347: The phrase "clearly ageing" confuses me a bit, since T' only ages in the period you defined as the RU heatwave. In fact, if you also took into account the days before July 31 (where it was already very hot), you could actually see deageing between July 30 and July 31.

L455-463: I think that is a very wise comment and it provides, in my opinion, a major reason why the research regarding heatwaves cannot be finished at this point!

**Technical corrections:**

L6: Replace "Western Russia" by "western Russian" as in L26 (or in L26 "western Russian" by "Western Russia")?

L60: Not totally sure, but I think you should replace "the stability of the atmospheric profile to moist convection" by "the stability of the atmosphere to moist convection" (since it's not the profile, which is stable/ unstable to convection, but the atmosphere itself). Same in L98 and 102.

L86: "the" in front of "Lagrangian analyses"?

L91: I find the word "transient" a bit confusing here. I would just drop this part of the sentence, since the following part of the sentence makes clear what you mean.

Fig 2/5/8/11: It's difficult to identify which of the date label corresponds to which tick on the x-axis. I suggest to rotate the date labels by 45°, such that they are aligned vertically.

L181: "19 July" instead of "20 July"?

L198: I suggest putting "(visible in the top left of Fig. 3a)" right after "was an upstream cyclone" (since it's the cyclone that is visible in the figure and not how it deepened rapidly).

Fig 3/6/9 (left column): The black and purple lines are really hard to see. Maybe you show less, but thicker lines to improve the visibility? Furthermore, I think the labeling with numbers is not needed at this point and dropping them may improves the visibility of the figure as well.

Fig 3 (caption): L1:Change "(i) the T' and its contribution" into "(i) the near-surface T' and its decomposition"?; L4: Drop "dashed" (or make the rectangle dashed in the figure); L7/8: What do you mean with "maximum 5-day daily T'" (same in L295)? I thought the gray lile denoted the period that you identified as your heatwave period. And I suggest putting the grey line all the way to the front (and not hiding it behind the light blue).

L225: "Causes" is a quite strong word. Maybe change "physical causes" into e.g. "physical (or underlying) processes"?

L232: "700" instead of "600"?

L241: Comma in front of Fig. S2?

---

## Author Comment (AC1)

**Reply to reviewer comments on "Understanding the vertical temperature structure of recent record-shattering heatwaves"**

By Belinda Hotz, Lukas Papritz and Matthias Röthlisberger

Institute for Atmospheric and Climate Science, ETH Zürich, Zürich, Switzerland

**General Comments to the reviewers:**

We would like to thank both reviewers for their thoughtful and positive reviews of our manuscript. In the revised version we will address all issues raised by the reviewers. Line numbers in our replies refer to those in the revised manuscript, unless stated otherwise. Reviewer comments are listed below in black font, while our replies appear in green.

**Reviewer 1**

**General comment:**

In this paper, the authors analyse the vertical temperature structure of three record-breaking heatwaves. In a first step, they use the diagnostics developed by Zhang and Boos (2023) to asses to what extent convective stability/instability played a role in determining the magnitude and duration of the heatwaves. In a second step, the authors then perform a detailed Lagrangian analysis based on the diagnostics developed in Röthlisberger and Papritz (2023). They quantify to what extend horizontal advection, vertical advection, and diabatic heating contributed to the specific events. By doing so, they go beyond many other Lagrangian- based studies in that they look at the entire vertical structure and not just the near ground. They find that the contribution from the individual processes varies significantly across the troposphere, with horizontal advection generally being a key process for establishing positive temperature anomalies in the mid to upper troposphere and adiabatic and diabatic warming in the lower troposphere; whereby large differences between the events have also occurred. Many of the results are consistent with the existing literature; in some cases the authors find contradictions. For sure, the analysis will stimulate future work in understanding the formation mechanisms of heatwaves. The manuscript is well-thought, well-written, and really worth reading. I especially enjoyed those parts that dealt with the Lagrangian analysis, which corresponds to the better part of the manuscript. Generally, I'm a fan of the Röthlisberger-Papritz-diagnostics and I'm convinced that its application brings us a good deal closer to understanding the underlying mechanisms in the development of temperature anomalies. Unfortunately, I had difficulties to follow the reasoning in Section 3.1, which deals with the role of convective instability, and how it relates to the (results of the) Lagrangian analysis. Below I have compiled a list of questions and comments, and I am eager to hear the authors' responses. If properly revised, I find the manuscript well suited for publication.

We thank the reviewer for the overall very positive assessment of our work and for the valuable comments – addressing these comments certainly helped to further improve the manuscript in terms of clarity! We have addressed all of your detailed comments and improved the clarity of Section 3.1.

**Major comments:**

1. Section 3.1: I have some difficulty following your reasoning in this section. My main problem is that I thought Zhang and Boos (2023) were arguing that a heat wave persists as long as the atmosphere is stable for moist convection, and that once convective instability is reached, precipitation sets in, ending the heat wave. However, for the PNW and the RU heatwaves you do see convective instability, but this does not end the heatwaves. It might help my understanding if you could describe what the plots in Figure 2 should look like to conclude that convective instability did not play a significant role in the heatwaves. Also, addressing my questions about L173-175 and L176/176 will certainly be helpful as well.

Thank you for this comment, which is important to clarify! We agree that Zhang and Boos (2023) argued that heat waves are preferentially ended when the atmosphere reaches moist convective neutrality concomitant with the onset of significant convective precipitation. In reverse, moist convective instability does certainly not limit surface temperatures if the stratification does not reach moist convective neutrality, as then there is no convective coupling between the surface and the free-tropospheric levels. Consequently, in such situations, no top-down control on near-surface temperatures in the sense of Zhang and Boos (2023) can be expected.

Following previous studies (e.g., Byrne, 2021; Zhang and Fueglistaler, 2020; Zhang and Boos, 2023), we quantify the convective coupling between surface and free tropospheric air by computing surface moist static energy (MSE_s) and saturation moist static energy at a free tropospheric level, here 500 hPa (MSE*_500). As in previous studies we identify a moist convectively neutral stratification if MSE_s=MSE*_500 and consider mid-tropospheric levels to be convectively coupled to surface air if MSE_s >= MSE*_500. Moist convective instability could thus be considered unimportant for limiting surface temperatures if the two red lines in Fig. 2 did not reach the same level.

For the PNW heatwave we see a good correspondence between the timing of the peak of the heatwave and the onset of significant precipitation, ending the heatwave. Interestingly, however, MSE_s–MSE*_500 indicates a moist convectively unstable atmosphere already 2-3 days earlier, yet the heatwave persisted further (but did not increase its intensity anymore). We find this observation "surprising" and remarkable and thus worthwhile documenting here. We suggest that the reason for this observation lies in the presence of substantial convective inhibition in the lower troposphere, which prevents the instability from being released. Similarly, for the RU heatwave we find MSE_s exceeding MSE*_500 during the hottest hours of many days. We hypothesize that in this case, the exceptionally dry conditions during the RU heatwave prevented significant precipitation and thus the heatwave did not decline in magnitude.

These findings suggest that near-surface and mid-tropospheric air were convectively coupled during the peak phases of at least the PNW and RU heatwaves (to a lesser extent during the UK heatwave). This in turn implies that, as for tropical temperature extremes (Byrne, 2021), mid- to upper-tropospheric temperatures played some role in limiting near-surface temperatures during the peak phases of these events. However, the fact that the RU heatwave persisted despite moist convectively unstable stratification and diurnal precipitation shows that other factors can modulate the role of convective instability for limiting/terminating a heatwave, such as the availability of moisture. The results of Zhang and Boos (2023), Byrne (2021) and others,

together with the finding of a moist convectively neutral or unstable stratification during the peak of our three events motivated us to study in detail the vertical structure of T' during the three heatwaves. Considering the Lagrangian T' decomposition as done in our study then unveils how this vertical T' structure formed, but we certainly agree that much is to be learned about the exact role of convection in extratropical heatwaves in further studies.

In response to your comment we have substantially reworded Section 3.1 (now lines 179-211) and hope that it is now much clearer.

2. L173-175: Here you write "Based on MSE*500-MSEs we find that during the PNW and RU heatwaves, the atmospheric vertical structure reached neutrality to moist convection during their peek phases. Hereby, negative values of MSE*500-MSEs are somewhat surprising." But I thought you were inferring from the negative values of the MSE*500 MSEs that the vertical structure achieved neutrality. So why are the negative values surprising if the atmospheric vertical structure has reached neutrality? My understanding is that it would be better if you wrote, e.g., "During the PNW and RU heatwaves, we find negative values of MSE*500 MSEs during their peak phases, indicating that the atmospheric vertical structure has achieved neutrality to moist convection at that time. This is somewhat surprising."

Negative values of $MSE^*\_500 - MSE\_s$ indicate convectively _unstable_ stratification, while a $MSE^*\_500 - MSE\_s = 0$ would indicate convective neutrality. We find it surprising that unstable conditions could be reached and, during some days, maintained for several hours, e.g., during the PNW and RU heatwaves. To clarify this point and in order to also account for a comment made by the second reviewer we now write on line 190–194: *"Hereby, negative values of MSE*\_500-MSE\_s (indicating unstable conditions) might have been sustained over multiple hours by entrainment of dry air into air parcels ascending through the PBL (as suggested by Zhang and Boos,2023), or by considerable convective inhibition (CIN), due to a strong boundary layer inversion (Fig.~S1a, Neal et al., 2022)."*

3. L176/176: You write that one should "note that [...] the near-surface temperatures peaked when negative [...] values started to appear". But I cannot see any line in Fig. 2 indicating temperature. I think this is also the reason why it's pretty hard for me to follow your reasoning in the whole section.

Indeed that formulation was somewhat unfortunate. We have removed this statement.

4. L266-270: Here, it's not clear to me why the pronounced diurnal cycle of the PBL height in combination with the constant vertical extent of the lower tropic diabatic T' shows that there is heat accumulation in the PBL. Could you explain this a bit more? E. g. What would the PBL height and diabatic T' look like if there had been no heat accumulation?

Important comment, thank you! If there was no diurnal accumulation of diabatically produced T', the diabatically produced T' in the boundary layer during a specific day would decay completely during the following night. In Fig. 5f, however, we see that some of the diabatically produced T' "survives" the night outside the PBL, and then re-enters the PBL as the PBL grows again during the course of the next day. This observation is at least qualitatively in accordance with the diurnal heat accumulation in the PBL proposed by Miralles et al. (2014).

To clarify this aspect the paragraph now reads (lines 285-290): *"Second, Fig.5f shows evidence of diurnal heat accumulation in the PBL, as observed in earlier mega heatwaves by Miralles et al. (2014). The positive diabatic T' in the PBL increases from 26 June to 30 June. While there is a diurnal cycle in the generation and decay of diabatic T', not all of the positive diabatic T' decays during the night, and some of it persists until the next morning, in particular in the residual layer above the PBL. On the following day, when the PBL grew again in depth, the air from the residual layer was likely again mixed into the PBL. Albeit not a definitive proof, these observations support the hypothesis of multi-day heat accumulation in the PBL in the sense of Miralles et al. (2014)."*

5. L279-281: As mentioned earlier, I'm not entirely convinced that you see evidence for a top-down control of T' via convective stability/instability, nor for the multi-day heat accumulation. But this probably changes if you clarify this a bit more in the earlier paragraphs.

Thank you, indeed the evidence for a top-down control needs to be presented and communicated clearly! We feel that in particular three observations support the notion of a top-down control of near surface T (or T') via convective instability: (1) Large T' started to appear first at upper levels and only later at lower levels. While this observation does not directly imply a top-down control, it is exactly the evolution of the vertical T' profile that is expected when a top-down control on near-surface T is indeed in place (Zhang and Boos, 2023). (2) The fact that values of MSE*_500 - MSE_s were close to zero (or even negative) indicates that the atmospheric profile above the PNW heat wave reached convective neutrality (or even instability), i.e., the mid-tropospheric atmosphere was convectively coupled to the boundary layer. In such a situation, additional heating at the surface can be expected to lead to convection (and thus damping of near surface temperatures). (3) Figure 2a shows a large precipitation peak co-occurring with the termination of the heat wave (1 July), which also coincides with a reduction in near-surface diabatic T' (Fig. 3i). This is exactly the signature expected in our analyses for a convectively terminated heat wave.

To clarify these three aspects we now explicitly refer to the MSE*_500 and MSE_s time series as well as the precipitation time series in Fig. 2. The sentence (on lines 301–304) now reads: *"In summary, the temporal evolution of the vertical T' structure together with the time series of precipitation, MSE*_500 and MSE_s support the notion of a top-down control on near-surface T' via convective (in)stability and provide qualitative evidence for multi-day diurnal heat accumulation in the PBL (as described in Miralles et al., 2014) preceding the peak of the heatwave, which also coincided with the time when strongly adiabatically warmed air was mixed into the PBL (Schumacher et al., 2022)."*

Regarding the multi-day heat accumulation during the PNW heatwave we hope to have clarified that (a) we see diabatic T' accumulating in the PBL, (b) some diabatic T' surviving the night in a layer above the PBL, which then re-enters the PBL upon PBL growth during the successive day. We are also very clear in the revised manuscript (lines 285-290) that our results are "consistent with" the hypothesis of multi-day heat accumulation and provide "qualitative evidence" in its favor – they do not, however, provide conclusive proof thereof (see comment above).

**Minor comments:**

6. Title: My feeling is that "understanding" is a very strong word that means to know the causes/reasons why things are the way they are. With your diagnostics, you can attribute a certain process to a certain T' and this is really great. But, is it enough to really understand the vertical temperature structure, i. e. to give a cause for it? In light of that (and your comment in L455-458), you might consider rephrasing the title a bit, e.g. into "Towards an understanding of the vertical temperature structure of ...". Or: "Disentangling the vertical temperature structure of ...".

   Interesting comment, thanks. We highly appreciate the subtle and philosophical nature of this comment and, knowing also the limitations of our work, we do understand how the reviewer came to formulate it. Nevertheless we would like to keep the original title of the manuscript and also feel it is justified to use the word "understanding" for the following reason:

   We believe that in many natural science endeavors finding "causes" is (i.e., answers to questions beginning with "why") is only possible within a predefined physical framework. In our case, the physical framework is rooted in the thermodynamic energy equation, which tells us that only three processes can form temperature anomalies in air parcels. Note that countless previous studies have at least conceptually used exactly the same physical framework and examined the same processes (advection, adiabatic warming/cooling and diabatic heating/cooling) in some form with the goal of "understanding" temperature extremes. We argue that in this commonly used framework of advective, adiabatic and diabatic contributions to temperature extremes our approach does provide an "understanding" of how T' formed at various levels during the three events. Given that there is quite a consensus in the scientific community about the usefulness of this framework for "explaining" temperature extremes, we do feel it is justified to claim that our study provides an "understanding" of how the vertical temperature structure of recent record shattering heat waves formed. But we do acknowledge that one could now go on to ask, why a certain T' composition appeared on a specific level etc. Clearly, though, such a next "why" question is always possible and the chain of "why" questions only ever ends if one defines the conceptual framework within which the "why" questions are to be addressed.

7. L23/24: I think what you do is that you analyse the large-scale dynamics (advection, subsidence, WCB), the moist convection (stability/instability), and a bit the boundary layer processes (PBL height), but at the moment I do not see that you really discuss their "interplay". If you think so too, consider rephrasing this sentence.

   Yes, we see that point. The sentence has been rephrased to (now on lines 22–24): *"Our results for the first time document the four-dimensional functioning of anticyclone–heatwave couplets in terms of advection, adiabatic and diabatic cooling or warming, and suggest that a complex interplay between large-scale dynamics, moist convection and boundary layer processes ultimately determines near-surface temperatures during heatwaves."*

8. L51: Which events do you mean with "events mentioned above"? The PNW, RU, and UK heatwaves or the "benign" heat waves from the previous paragraph?

   Here, we meant here our extreme heatwave case studies. The sentence has been rephrased to

(now lines 50–54): *"Case studies focusing on exceptionally intense heat waves such as the events mentioned above have confirmed ..."*

9. L91/125: Here, I'm interested in a technical detail. If you compute the mean T on modellevels, what do you then take as $\Delta p$ in the computation of the vertical gradient? E. g. do you compute a climatological mean pressure for each modellevel at each grid point, from which you then infer the $\Delta p$ between each modellevel at each grid point? Or do you take the instantaneous values of p?

We use the instantaneous values of the surface pressure PS to determine p on model-level. The vertical derivative of T_clim at model level k we then compute as finite centered difference using the T_clim and p values on levels k+1 and k-1. For the lowest and highest model level we set the vertical gradient of T_clim to zero. That is, we essentially handle T_clim just as any other model-level variable.

10. L93: I do not fully understand what you mean by "9-year windows centered on the time step of interest". Especially I was wondering how you handle dates at the edges of the ERA5 timeseries when there are no 4 years left, e. g., dates in the year 2021 or 2022. Could you clarify this?

Yes, we hope the following example helps: The T_clim values at, say, 12 UTC 20 January 2000 are computed by taking the average over all 12 UTC time steps on all days between (and including) 10 January to 30 January, from the years 1996–2004. As in previous publications (e.g., Röthlisberger and Papritz, 2023a,b) we use the last (first) 9 years to compute the T climatology in the last (first) five years of the ERA5 period (1979 to 2022 in this study). Note that this implies that for the PNW and UK heat waves we used the years 2014 to 2022 to compute the T climatology. Since we are interested in temperature extremes here (i.e., very large temperature anomalies), the exact details of how the T climatology is computed are not particularly relevant, as long as the climatology includes both the climatological diurnal and seasonal cycles. We have added an example analogous to the one above and a clearer explanation on lines 94–97.

11. L121: Can the LAGRANTO trajectories account for turbulent mixing in the PBL?

For computing the trajectories we use the full wind fields, also in the PBL, which due to the 0.5° resolution cannot explicitly represent turbulent mixing. While the mean winds are also influenced by momentum tendencies from the turbulence scheme, LAGRANTO does not explicitly account for turbulent mixing such as done for certain particle dispersion models (e.g., FLEXPART, Stohl et al. 2005). Thus, the air parcels move with the mean wind. Any sub-gridscale turbulent mixing then shows up as a diabatic tendency, i.e., as changes of potential temperature along a trajectory. However, it is clear that with 0.5° resolution, small scale turbulence is not resolved in ERA5. To improve the robustness of our analyses we have very deliberately computed trajectories from three near-surface levels (i.e., 10, 30 and 50 hPa above ground). In section 2.3 we now explicitly point to this caveat of LAGRANTO/ERA5 (now lines 124–126).

12. L125: I was just wondering why you trace specific humidity q. I suppose just in case it's needed at some point? If so, you could drop q in this listing here to prevent confusion.

Well spotted! We did trace q for an analysis which we chose not to present here in the paper. We have removed the q.

13. L160/161: Do the regions and periods of the other two heatwaves also correspond to the event definitions in other studies?

Yes. For the RU heatwave the box used here is very similar in size and location to event definitions used in Dole et al. (2011) and Hauser et al. (2016), and the selected heat wave period covers the peak of the heat wave, as also indicated by these two studies. For the UK heatwave few studies focusing on this event are available, but our heat wave period contains the day when temperatures exceeding 40°C were first measured in the UK (19 July 2022, e.g., Yule et al. 2023). We have rephrased the respective sentence (now lines 164–167).

14. L166/167: This goes back to my previous comment. You write that your results are "insensitive to horizontal shifts of the heatwave regions by a few degrees, enlarging or shrinking the regions by a few degrees, and shifts of the heatwave periods by 1-2 days." For the PNW and the UK heatwaves, I can well imagine that this is true. However, for the RU heatwave, which was quite extensive spatially and lasted over a month in total, your results may not be representative of the entire heat wave period/region. So it could well be that a shift of a few days doesn't matter, but a shift of, say, 10 days might actually matter and your results would have looked different if, e. g., you had chosen a period closer to the onset phase of the heatwave. I think you should mention this a bit more prominently (see also my comment to L347)

True. We have added the following sentence on lines 174–176: *"Note however, that the RU heatwave lasted for over a month (Barriopedro et al., 2011). Here, we focused on its peak phase, thus we cannot exclude the possibility our results may be sensitive to much larger shifts of the respective heatwave period."*

15. L206: In Fig. 3b, I can't see that the air parcels first ascended and then descended. I think this is because the trajectories overlap and cover each other and only a small part of the colorbar is used. I think the figure would improve if you adjusted the colorbar accordingly and made the trajectories a little transparent. Maybe it would also be a good idea to take trajectories that end up more apart. The same suggestions apply to Fig. 6 and 9.

Thank you for this comment. We see your difficulty in identifying the ascent and descent of the air masses along the trajectories in Fig. 3b. We will try to enhance the clarity of the second column of Figs. 3, 6 and 9 by adapting the colormap and reducing the overlapping trajectories in the revised manuscript.

16. L254: Determining the Lagrangian age and formation distances would be possible without doing the full T' decomposition. Thus, reformulating this sentence into e.g. "By determining the Lagrangian age and formation distances, the temporal and spatial scales over which the temperature anomalies form can be quantified." might be more precise.

Thank you for spotting this. It is true that it is not necessary to fully decompose the temperature anomalies for determining the Lagrangian age and formation distance since we only use the instantaneous and the climatological temperatures along the trajectories to calculate these quantities. We have rephrased this sentence according to your suggestion (now lines 272–275).

17. L283 (and L21 and L453): Here you argue that the aging of T' suggests the concept of a "heat dome" in which "air recirculates and accumulates heat". I agree with you that a heat dome would have exactly these properties, i.e., aging of T', and that it is plausible to observe this during the PNW heatwave. However, the aging of T' is not conclusive evidence that the air masses are actually recirculating. What can "only" be seen is that the air masses that happen to arrive in the heat wave region tend to be older (in terms of temperature anomaly lifetime) than before. But you can't be sure if it's always the same air masses you observe, or if it's new air masses that happen to have an older lifetime. If you agree, I would appreciate a comment on this when you make statements about the "heat dome".

We agree. we have made the following changes:
Lines 20–21 (abstract): We now write *"... the widespread "ageing" of near-surface T' over the course of these events is fully consistent with the notion of heat domes ..."*

Lines 475–479 (conclusions): *"Finally, for all three cases, we find evidence of ageing of T', in particular for lower-tropospheric air that subsided significantly before contributing to its respective heatwave. This result is fully consistent with the concept of a "heat dome", within which air re-circulates and accumulates heat. However the ageing of T' does not directly imply re-circulation of air, but simply shows that the anomalies contributing to our events of interest became older throughout the course of the events."*

On the original line 283 we think everything is in order, as we don't imply there that the "ageing" we see is related to the heat dome. We just write there (now lines 305–306) that *"... for the first time, documents the ageing of T' throughout the PNW heatwave, which has been qualitatively surmised by previous studies putting forward the concept of a "heat dome" (Neal et al., 2022, Zhang et al., 2023)."*

18. Fig. 5/8/11c: Never mentioned/discussed in the text.

Yes, indeed these panels are not in the center of our discussions. We nevertheless find them insightful and would like to keep them in these figures. Note that Fig. 11c, was already mentioned in L385 of the original paper (now L409). For the other two subplots, we now also mention Figs. 5c and 8c on lines L284 and L355.

19. L347: The phrase "clearly ageing" confuses me a bit, since T' only ages in the period you defined as the RU heatwave. In fact, if you also took into account the days before July 31 (where it was already very hot), you could actually see deageing between July 30 and July 31.

Well spotted. We have specified more clearly to which period we refer and now write (line 371–373): *"Finally, similarly to the PNW heatwave, also during the RU heatwave T' in the lower troposphere were ageing (Fig. 8b) during the peak phase of this event."*

20. L455-463: I think that is a very wise comment and it provides, in my opinion, a major reason why the research regarding heatwaves cannot be finished at this point!

Thank you, we couldn't agree more! Our study is not the end of the story, but hopefully it will prove useful and stimulating to our colleagues around the globe who are working on similar topics!

**Technical corrections:**

21. L6: Replace "Western Russia" by "western Russian" as in L26 (or in L26 "western Russian" by "Western Russia")?

    Thank you, we capitalized it on line 26.

22. L60: Not totally sure, but I think you should replace "the stability of the atmospheric profile to moist convection" by "the stability of the atmosphere to moist convection" (since it's not the profile, which is stable/ unstable to convection, but the atmosphere itself). Same in L98 and 102.

    We agree and changed "the stability of the atmospheric profile" to "the stability of the atmosphere" on lines 61, 101 and 105.

23. L86: "the" in front of "Lagrangian analyses"?

    Thank you, changed!

24. L91: I find the word "transient" a bit confusing here. I would just drop this part of the sentence, since the following part of the sentence makes clear what you mean.

    Ok, changed.

25. Fig 2/5/8/11: It's difficult to identify which of the date label corresponds to which tick on the x-axis. I suggest to rotate the date labels by 45°, such that they are aligned vertically.

    Thank you for your suggestion. We agree that the readability of the dates could be improved and have adapted the x-axis labels of Figs 2, 5, 8, 11, S1 and S2 in the revised manuscript.

26. L181: "19 July" instead of "20 July"?

    Thank you for pointing this out. Yes, it should be the 19 July. The previously suggested adjustment of the x-axis helps readability in this case as well. We have adjusted these labels in the revised manuscript.

27. L198: I suggest putting "(visible in the top left of Fig. 3a)" right after "was an upstream cyclone" (since it's the cyclone that is visible in the figure and not how it deepened rapidly).

    Ok, thanks, your suggestion helps avoid misunderstandings. We adopted your suggestion.

28. Fig 3/6/9 (left column): The black and purple lines are really hard to see. Maybe you show less, but thicker lines to improve the visibility? Furthermore, I think the labeling with numbers is not needed at this point and dropping them may improves the visibility of the figure as well.

Thank you for your comment. We agree that the visibility in the left panels of Fig. 3/6/9 should be improved. We dropped the inline labels and will try to further improve the visibility within the panels for the revised manuscript.

29. Fig 3 (caption): L1:Change "(i) the T' and its contribution" into "(i) the near-surface T' and its decomposition"?; L4: Drop "dashed" (or make the rectangle dashed in the figure); L7/8: What do you mean with "maximum 5-day daily T'" (same in L295)? I thought the gray lile denoted the period that you identified as your heatwave period. And I suggest putting the grey line all the way to the front (and not hiding it behind the light blue).

Thank you for your comment and your suggestions about this caption. We changed the "dashed black rectangle" to "black rectangle". Moreover, we adapted "(i) the T' and its contribution" to "(i) the near-surface T' and its decomposition" in order to clarify the type of the temperature anomaly. Further, we moved the gray line in panel (i) to the front. We also applied these changes to Figs. 6 and 9. Our identified heatwave period, indicated with the gray line in panel (i) corresponds to the five-day period with maximum 5-day average near-surface T' in our case study region (black rectangles). The last sentence in the caption to Fig. 3 now reads: *"The grey line denotes the PNW heat wave period."*

30. L225: "Causes" is a quite strong word. Maybe change "physical causes" into e.g. "physical (or underlying) processes"?

We find "causes" justified in this particular statement. After all, we do not just mean the terms in our decomposition here but also their further interpretation in terms of weather systems (e.g., the WCB "causing" positive T' in the mid-to upper troposphere above the PNW heat wave).

31. L232: "700" instead of "600"?

Thanks, corrected!

32. L241: Comma in front of Fig. S2?

We agree, corrected!

**Reviewer 2**

**General comment:**
Drawing inspiration from recent advancements in understanding atmospheric stability and stratification in heatwaves, and leveraging the Lagrangian framework developed by Roethlisberger and Papritz (2023), this paper delivers the first meticulous Lagrangian analysis of the 3-D atmosphere during three extreme heatwaves. This study is technically robust and discusses properly the existing body of literature. It provides a fresh perspective on heatwaves and holds the potential to expand its findings to analyze more events in future research. I believe this manuscript is deserving of publication following some minor revisions.

Thank you for the overall very positive assessment of our work as well as your comments below!

**Minor comments:**

1. Line 174: The negative values are likely due to strong entrainment, which dilutes the specific humidity of air parcels as they ascend through the dry boundary layer, as noted by Zhang and Boos.

   Thank you, we now explicitly refer to this hypothesis put forward by Zhang and Boos (2023) on lines 191–192.

2. Line 215: I'm not entirely convinced that the conclusion of the PNW heatwave was primarily due to convective damping rather than changes in advection. Utilizing an Eulerian viewpoint might offer a clearer understanding of the decline in boundary layer temperature. If the Eulerian advection term continues to rise when the surface temperature is already declining, then it is reasonable to infer that advection is not the primary cause. Additionally, the T' values utilized here represent accumulations, not tendencies. Thus, discussions regarding changes in total T' should center around the changes in advective T'. I am not requesting additional analysis given the Lagrangian focus of this paper, but anticipate that the authors will address these reservations and refine their discussion accordingly.

   We agree that the cause of the PNW heat wave termination is indeed not entirely clear cut from the analyses presented in Figs. 2 and 3. However, given the focus on Lagrangian diagnostics in this study we prefer to not include an additional, Eulerian T' budget just to underline this one aspect of the PNW heat wave. While we still believe our initial statement was correct (after all the precipitation peak does coincide with the termination of the heat wave, while little change in the advective T' excludes the possibility that air from climatologically colder regions enters the PNW region at this time), we have now toned this statement down to some degree (lines 235–236): *"This is consistent with the termination of the PNW heatwave being due to convective damping of near-surface T' rather than due to changes in air mass advection into the region."*

3. Line 271: This paragraph is somewhat unclear to me. It appears that by 29 June, the peak adiabatic T' has already reached 850 hPa, well below the PBL top. My interpretation is that the positive adiabatic T', rather than the peak, only descended to the PBL top around 29-30 June. Could the authors clarify whether they intend to convey that the peak of the surface temperature

aligns with the time when adiabatically heated air stopped to mix into the boundary layer?

*We here wanted to refer to a hypothesis put forward by Schumacher et al. (2022), who suggested that mixing of adiabatically heated air into the PBL upon diurnal PBL growth contributed to the exceptional near-surface heat. We now explicitly refer to Schumacher et al. (2022) in this sentence, and have re-written this paragraph. It now reads (lines 291–296) "This is consistent with the hypothesis that during the peak days of the PNW heat wave, some of the air that was mixed into the PBL during the diurnal PBL growth had been significantly heated adiabatically before (Schumacher et al., 2022), although a vertical propagation of signals in Fig. 5 over time does not necessarily imply that the same air parcels contribute to that signal at different time steps."*

4. Line 325: Might the RU heatwave exhibit more similarities to the PNW heatwave if evaluated over a longer time frame? There is no need for the authors to change the figures, but a brief comment on this would be beneficial.

*True. We have added the following sentence on line 174–176 (Section 2.5): "Note however, that the RU heatwave lasted for over a month (Barriopedro et al., 2011). Here, we focused on its peak phase, thus we cannot exclude the possibility our results may be sensitive to much larger shifts of the respective heatwave period."*

**References:**

Byrne, M.P. Amplified warming of extreme temperatures over tropical land. *Nat. Geosci.* **14**, 837–841 (2021). https://doi.org/10.1038/s41561-021-00828-8

Stohl, A., Forster, C., Frank, A., Seibert, P., and Wotawa, G.: Technical note: The Lagrangian particle dispersion model FLEXPART version 6.2, *Atmos. Chem. Phys.*, 5, 2461–2474 (2005) https://doi.org/10.5194/acp-5-2461-2005.

Zhang, Y., & Fueglistaler, S. (2020). How tropical convection couples high moist static energy over land and ocean. *Geophys. Res. Lett.* 47, e2019GL086387, (2020). https://doi.org/10.1029/2019GL086387

---

## Referee Report (RR1)

**Second review of "Understanding the vertical temperature structure of recent record-shattering heatwaves" by B. Hotz, L. Papritz and M. Röthlisberger submitted to Weather and Climate Dynamics**

**General comment:**

I thank the authors for their careful consideration of all my comments! Especially, Section 3.1 and the discussion relating to the diurnal heat accumulation in the PBL are now clear to me. I'm looking forward to the publication of the manuscript.

**Technical correction:**

L: 194: Didn't $MSE_s$ peak on 19 July rather than on 20 July?

---

## Author Response (AR2)

**Second review of "Understanding the vertical temperature structure of recent record-shattering heatwaves" by B. Hotz, L. Papritz and M. Röthlisberger submitted to Weather and Climate Dynamics**

**General Comments to the reviewers:**

We acknowledge both reviewers for taking the time and effort in reviewing our manuscript. Reviewer comments are listed below in black font, while our replies are provided in green.

**Reviewer 1:**
**General comment:**
I thank the authors for their careful consideration of all my comments! Especially, Section 3.1 and the discussion relating to the diurnal heat accumulation in the PBL are now clear to me. I'm looking forward to the publication of the manuscript.

We would like to thank the reviewer again for the useful feedback and suggestions for improving our manuscript.

**Technical correction:**
1. L194: Didn't $MSE_s$ peak on 19 July rather than on 20 July?

   Thank you for pointing this out. Yes, it should be the 19th of July. We have rectified it in the revised manuscript.